

# Opinion: how are advances in aerosol science informing our understanding of the health impacts of outdoor particulate pollution?

Imad El Haddad[1]*, Kaspar Daellenbach[1], Robin Modini[1], Jay Slowik[1], Abhishek Upadhyay[1],
David Bell[1], Danielle Vienneau[2, 3], Kees De Hoogh[2, 3]*, and Andre S.H. Prevot[1]

[1] Laboratory of Atmospheric Chemistry, Paul Scherrer Institute (PSI), 5232 Villigen, Switzerland
[2] Swiss Tropical and Public Health Institute, Basel, Switzerland
[3] University of Basel, Basel, Switzerland

*corresponding authors: imad.el-haddad@psi.ch and c.dehoogh@swisstph.ch

**Abstract.** Air pollution poses the greatest environmental threat to human health, causing an estimated nine million premature deaths annually and accounting for 5% of the global GDP. This opinion paper explores how advances in aerosol science inform our understanding of the health impacts of outdoor particulate pollution. In the article, we advocate for a shift from solely considering total particulate matter (PM) mass to utilizing specific PM components as metrics for health assessments. This will allow targeted evidence-based interventions, limiting the most harmful anthropogenic emissions, while exempting uncontrollable or non-detrimental components from guidelines. Central to this shift is the availability of global long-term PM chemical composition data obtained through field observations and modelling outputs. These data will serve as the new foundation for identifying the most harmful chemical components in different regions. We discuss emerging modelling tools for personalized exposure estimation to these components, present the type of ambient observations needed for model evaluation and highlight key gaps in our fundamental understanding of emissions and their health effects. Through global PM chemical composition data, advancements in modelling tools, and collaboration between aerosol scientists and epidemiologists, we can gain a causal understanding of how different PM components influence disease development. The reevaluation of air quality guidelines with a focus on specific PM components will be essential for fostering healthier environments, preventing diseases and building resilient communities.

## 1. Preamble

### 1.1 A brief chronology of air pollution

A tale of global air pollution has already been narrated by Fowler et al., and only a brief chronology will follow, presenting the main milestones reached by the atmospheric science community since the earliest recorded accounts of air pollution (Fowler et al., 2020). The threat of air pollution to human health has been recognized since the time of Hippocrates, about 400 before our era (Jones et al., 1923). Successive written accounts of air pollution occur throughout the following two millennia until measurements from the eighteenth century onwards demonstrated the growing scale of poor air quality in urban centres. One of the most emblematic early historical documents on air pollution was published in 1661 under the title *Fumifugium* by Evelyn (Evelyn, 1772). Evelyn documented the air pollution in London and proposed solutions for reducing the scale of the problem by moving industries to the countryside. Graunt, a contemporary of Evelyn, observed a correlation between rates of mortality and pollution, especially in fog episodes, albeit in the absence of any chemical data or numerical values to quantify the pollutants present (Graunt, 1939). Later, in 1775, Sir Percival Pott was one of the first to document the effects of specific pollutants on health. Pott observed a high incidence of scrotal cancer



among chimneysweepers and concluded that exposure to soot was a risk factor for the cancer (Brown
and Thornton, 1957).

44        The industrial revolution accelerated the growth and geographical spread of emissions, as
highly polluted cities became the defining problem that culminated with the great smog of London in
1952. This pollution episode of a few days duration caused an estimated death of 10,000 persons and
the injury of more than 100,000 (Stone, 2002; Bell et al., 2004). London's smog is believed to be the
worst air pollution event in the history of the United Kingdom and the most notorious for its effects on
environmental research, government regulation, and public awareness of the relationship between air
quality and health. It was instrumental for establishing an unambiguous link between short-term
exposure to peak levels of pollution and acute health effects. It also led to the introduction of the Clean
Air Act of 1956 that aimed to reduce emissions and mitigate future pollution events. Until the latter
decades of the twentieth century, Europe and North America dominated global emissions and suffered
the majority of adverse health and environmental effects. By that time, the transboundary issues of acid
rain (Egnér and Eriksson, 1955) and ground-level ozone (Volz and Kley, 1988; Fowler et al., 2008)
were the focal environmental and political air quality problems (Vasseur, 1973). As emission controls
began to take effect in the West, pollution worsened in Asia due to its rapid industrialization, eventually
becoming the dominant source of global emissions by the early years of the twenty-first century.

59        Towards the end of the 20[th] century, the health effects of air pollution resurfaced as a top
priority, as new epidemiological evidence highlighted the breadth of chronic health problems resulting
from long-term exposure to relatively low levels of pollution (Dockery et al., 1993). For this, the
emergence of extensive networks of surface measurements, satellite remote sensing, and numerical
models was indispensable for providing global air quality data with which epidemiologists could
estimate the adverse health effects of air pollution. Since then, numerous studies have documented the
chronic and acute health effects of air pollution, many with a global perspective (Burnett et al., 2018;
Cohen et al., 2017; Mcduffie et al., 2021; Richard T. Burnett, 2014; Lelieveld et al., 2015; Lelieveld et
al., 2019; Chen et al., 2018b; Chen and Hoek, 2020; De Bont et al., 2022; Holtjer et al., 2023; Nyadanu
et al., 2022). Today, air pollution remains a major public health concern, and efforts continue to reduce
emissions and improve air quality.
**1.2 Particulate air pollution**

71        The polluted air we breathe contains high levels of particulate matter, PM, commonly termed
aerosols. PM is a complex mixture of tiny solid or liquid particles suspended in the air, with a size
ranging from few nanometers to few micrometers (John H. Seinfeld, 2016). These particles can be
directly emitted from primary sources, e.g. desert dust or soot from combustion emissions. They can
also be formed in the atmosphere by gas-to-particle conversion of secondary oxidation products, e.g.
sulfate from $SO_2$ oxidation, nitrate from $NO_X$ oxidation or secondary organic aerosol (SOA) from the
oxidation of volatile organic vapours. PM sources can be either natural or human-made. Natural sources
include desert dust, sea-spray, wildfires and biogenic SOA from the oxidation of plant volatiles, while
anthropogenic sources include emissions from residential heating or car exhaust and their secondary
oxidation products. As a result, PM has an immensely complex chemical composition with different
levels of toxicity depending on the emission sources and/or formation processes (Hallquist et al., 2009;
Jimenez et al., 2009). Smaller particles are more likely to enter our bloodstream and travel deep into
our lungs, causing damage. Short-term exposure to peak levels of PM, akin to the great London smog
of 1952, can cause acute health effects. By contrast, long-term exposure to low PM levels leads to
chronic diseases, such as cardiovascular (De Bont et al., 2022), cerebrovascular and respiratory diseases



(Holtjer et al., 2023), which are responsible for most of the estimated air-pollution-related mortality
(Burnett et al., 2018; Cohen et al., 2017; Chen and Hoek, 2020). Current epidemiological evidence
reveals that no level of air pollution can be deemed safe and even low levels of PM may carry significant
risks (Strak et al., 2021; Pinault et al., 2016; Cohen et al., 2017; Dominici et al., 2022; Brunekreef,
2021; Brauer et al., 2019). Today, PM pollution is responsible for nine million deaths every year
(Burnett et al., 2018). It classifies among the five leading causes of premature deaths worldwide,
alongside with high blood pressure, smoking, diabetes and obesity (Cohen et al., 2017).
**1.3 PM mitigation: a global challenge of the 21st century**
Although particles are compositionally heterogeneous, showing marked temporal and spatial
variations, most studies investigating their adverse health effects tend to treat them as a uniform entity,
summarised by a mass concentration in the air. Consequently, particle mass concentration, primarily
$PM_{2.5}$ in the USA and $PM_{10}$ in Europe[1], was routinely measured and formed the basis of epidemiological
observations connecting exposures to air pollution with health records at the population level. As a
result, PM mass serves today as the primary metric for particulate pollution regulation.
In response to the mounting evidence of the negative health effects of PM, the World Health
Organization, WHO, has recently updated its air quality guidelines to propose a much more stringent
limit value of 5 µg m$^{-3}$ (Who). These new guidelines provide a basis to justify aggressive regulations of
anthropogenic emissions in order to improve global air quality. Such low PM concentrations are
currently only found in some remote environments, while over 95% of the world population lives in
places where the new guidelines are not met. Several western countries have made significant progress
over the past 20 years in order to meet the former WHO limit of 10 µg m$^{-3}$ last updated in 2005
(Southerland et al., 2022; Hammer et al., 2020). In contrast, PM levels exceeding 50 µg m$^{-3}$ are typical
in low- to middle-income countries, e.g. in Eastern-Europe, China or India, where 90% of PM-related
deaths occur (Lelieveld et al., 2015). This translates to a loss of several years of life expectancy in Asia
due to pollution, compared to several months in the West (Lelieveld et al., 2019).
Reducing fossil fuel and residential emissions will undoubtedly significantly improve air
quality, especially in polluted environments (Pai et al., 2022; Mcduffie et al., 2021). However, natural
sources including desert dust, wildfires and biogenic emissions will impede many regions from
complying with the new WHO guidelines. A recent landmark modelling analysis suggests that over
50% of the global population will still be living in places with $PM_{2.5}$ concentrations greater than 5 µg
m$^{-3}$, even if all anthropogenic emissions would be eliminated (Pai et al., 2022). Moreover, natural
emissions are likely to increase in the near future, further complicating efforts to meet the new WHO
guidelines in certain regions (Gomez et al., 2023). Meeting these guidelines will be particularly
challenging for many regions worldwide, and globally applicable solutions to manage and improve air
quality will become no longer evident. This entails a complete rethink of how we should be mitigating
air pollution and suggests a need for a new generation of feasible air quality metrics that focus on
specific anthropogenic PM components in addition to total PM mass.
Another benefit in targeting particulate pollution across individual chemical components is that
different components have varying toxicity. This is termed the differential toxicity of PM components
(Masselot et al., 2022). Epidemiological analyses of PM health effects, which constitute the foundation
for mitigation strategies, have been based on total PM mass concentrations, which are readily available
globally through in-situ measurements and remote sensing. However, PM health effects are mediated

---

[1] $PM_{2.5}$ and $PM_{10}$: Particulate matter with a size lower than 2.5 and 10 µm.



by their size, solubility and chemical composition, and hence their sources and formation processes. In
our recent work, we have identified the organic and metal fractions to be of particular concern for
oxidative stress (Daellenbach et al., 2020) and inflammation (Leni et al., 2020), in contrast to secondary
inorganic particles that dominate PM mass. Given the role of oxidative stress as a major driver of PM
health effects (Mudway et al., 2020), this necessitate a reconsideration of which sources of PM should
be mitigated. It is vitally important that atmospheric scientists provide policymakers with global PM
chemical composition data, which will constitute a new basis for identifying the most harmful chemical
components, enabling targeted cost-effective decision-making for limiting specific health-relevant
anthropogenic PM sources in different regions.

### 1.4 Understanding the health effects of PM constituents

This perspective article discusses how the broader atmospheric science community can help
informing strategies aimed at reducing the sources of PM components that pose the greatest risks to
human health (Figure 1). The article introduces the concept of using specific PM components as metrics
for health assessments in addition to total PM mass. We will present new advances in modelling tools
that enables the estimation of personalized exposures to these components. We will then discuss which
ambient observations are necessary for model validations and address the gaps in our understanding of
PM emissions and their health effects. Lastly, we will discuss novel epidemiological data needed to
gain insights into the biological mechanisms underlying the impacts of these PM components on our
health. The article holistically addresses the critical aspects of the PM pollution field, presenting key
observations and developments needed, in our opinion, to shift the focus towards quantifying the health
impacts of individual PM components.

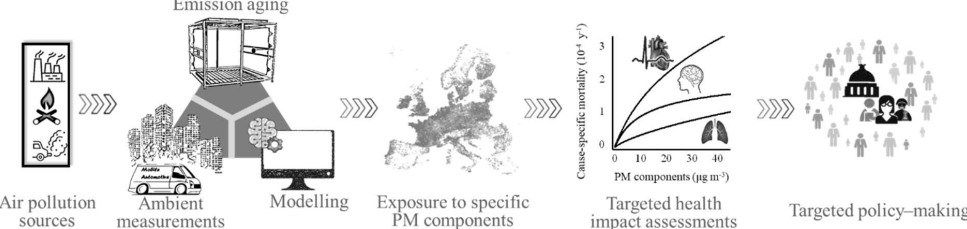

**Figure 1:** A multidisciplinary framework for the identification of the health relevant PM components.

## 2. PM air quality data relevant for health impact assessments

### 2.1 Targeted PM air quality metrics: more than just PM mass

To quantify the health impacts of PM, we currently rely on dose-response relationships that link
cause-specific mortality to the concentration of total PM mass ideally utilizing individual-level data
from large cohort studies. Whilst these relationships are consistent across studies, there is significant
heterogeneity in the estimated effect size among them. This variation can be partially attributed to
imperfect models approximating individual exposures or random differences among study populations.
Yet, perhaps the largest source of error lies in relying solely on PM mass concentration, ignoring the
biological activity of different particle constituents and leaving us unaware of the causal pathways that
link the complex chemistry of the air we breathe to disease development. Although some studies have
attempted to examine the adverse health outcomes of PM components, particularly highlighting
associations with combustion and road traffic emissions, such investigations remain relatively
infrequent.



With the advent of vast amounts of atmospheric data, the time has now come to redirect our focus towards developing dose-response relationships that describe the specific health effects of individual PM constituents rather than the more general quantity of total PM mass. In practical terms, these constituents must be quantifiable, easily accessible and readily available at high resolution and large spatial scales. Our proposal includes considering the following constituents: organic aerosol, elemental carbon, sulfate, nitrate, ammonium, sea-salt, brake-wear and dust. While brake-wear and dust concentrations cannot be directly measured, they can be traced using specific markers, such as Cu for brake-wear and Al for dust. The organic fraction should be ideally subdivided into several classes, each related to a distinct source sector, including primary and secondary aerosols from car exhaust, residential burning, wildfires and biogenic emissions. While organic aerosol classes cannot be directly measured, they might be retrieved through receptor modelling based on spectrometric measurements or chemical transport modelling, as discussed below. The classification of aerosols based on their chemical composition not only elucidates the causal connections between exposures and health risks, but also establishes a direct link to aerosol sources, offering an effective strategy for mitigating the most important sources for health.

Beyond PM chemical composition, other properties have been proposed to mediate different aerosol health effects, including aerosol size, number, solubility and oxidative potential. For example, toxic metals can cause oxidative damage mainly when they are in their soluble form (Fang et al., 2017; Wong et al., 2020), whereas insoluble particles, such as asbestos or elemental carbon, can bio-accumulate and lead to chronic inflammation. Likewise, small particles can penetrate deep into the lungs, enter the bloodstream and cross the blood-brain barrier causing respiratory, cardiovascular and neurological diseases (Requia et al., 2017; Maher et al., 2016), while significant fraction of large particles is ingested causing an imbalance in our gut microbiome (Fouladi et al., 2020; Alderete et al., 2018; Bailey et al., 2020). Parameters for emerging metrics intended to be used in future epidemiological studies should be standardized and widely available. PM chemical composition is intertwined with these alternative metrics, and therefore we argue that targeting PM based on its chemical composition is the most effective approach to address PM health impacts.

## 2.2 Necessity of fine-resolution pollution data for exposure assessments

The most polluted environments are in densely populated urban agglomerations (Mcduffie et al., 2021) and 70% of the world population is projected to live in urban areas by 2050. The composition and concentrations of PM in these areas exhibit significant spatial heterogeneity on street to citywide scales. In some cases, intra-city variability exceeds the variability between different cities (De Hoogh et al., 2016; Eeftens et al., 2016; Tsai et al., 2015; De Hoogh et al., 2013; Eeftens et al., 2012a; Zhang et al., 2015; Jedynska et al., 2015). Such spatial heterogeneity is driven by traffic patterns (Simon et al., 2017; Li et al., 2016; Gu et al., 2018; Elser et al., 2018; Elser et al., 2016), restaurant emissions (Gu et al., 2018), domestic heating emissions (Elser et al., 2018; Elser et al., 2016; Jedynska et al., 2015; Mohr et al., 2011), industrial point sources (Shairsingh et al., 2018) and local geography (Mohr et al., 2011). Atmospheric aging of urban emissions and long-range transport of polluted air masses add to this complexity, affecting PM background levels, composition and health effects on regional scales.

Urban microenvironments strongly affect long-term exposures to several PM components (Figure 2A). For example, there is a strong link between road proximity, exposure to ultrafine particles, and respiratory, cardiovascular and neurodegenerative diseases (Alexeeff et al., 2018; Bayer-Oglesby et al., 2006; Yuchi et al., 2020; Boogaard et al., 2022). It has also been shown that exposures to high particle concentrations around train stations during typical daily commutes of less than one hour can



contribute up to 21% of total daily PM exposure and more than 50% of daily exposure to toxic metals
such as Cu (Van Ryswyk et al., 2017). Therefore, the knowledge of PM chemical composition on fine
spatial scales relevant to daily human activities is imperative for assessing human exposures to specific
PM components.

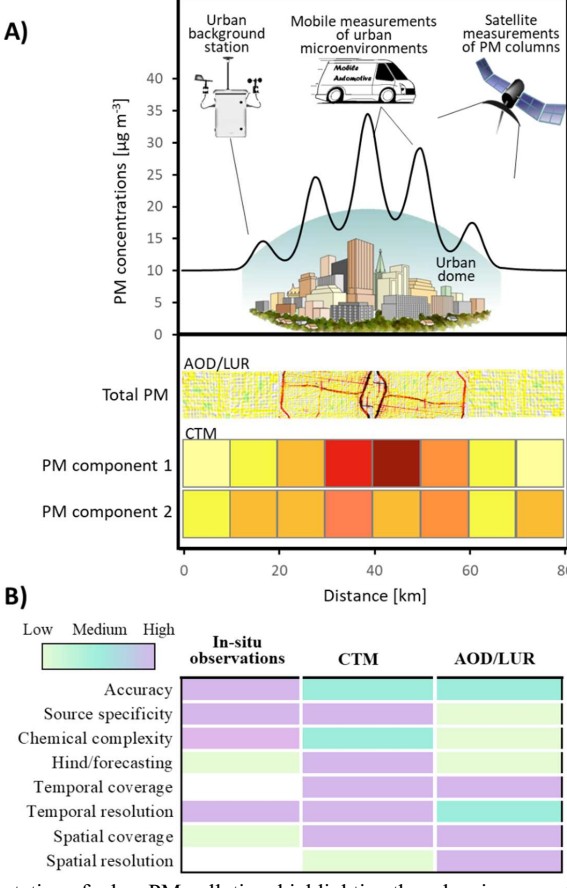

**Figure 2: A)** Representation of urban PM pollution, highlighting the urban increments in PM concentrations
over background levels and the presence of microenvironments. State-of-the-art measurement and modelling
strategies of PM concentrations at different scales are presented and compared in **B)** in terms of their
advantages and limitations. Three different approaches are compared including field observations, chemical
transport modelling (CTM) and land-use regression models based on aerosol optical depth (AOD/LUR). The
temporal coverage and spatial resolution of in-situ observations are determined by the method employed to
obtain them, with white cells being assigned accordingly. Comparison of the performance of CTM vs.
AOD/LUR is illustrated in **A)**, showing the source specificity of CTM and the high resolution of the
AOD/LUR.
In most epidemiological analyses, human exposures are typically based on outdoor PM
concentrations estimated at the residence place. However, since we spend the majority of our times
indoors and new buildings are increasingly airtight for energy saving, outdoor air pollution may not
accurately reflect individual exposures (Schweizer et al., 2007). While indoor air pollution, primarily
from cooking (Klein et al., 2019) and smoking (Hyland et al., 2008), may pose significant concerns, it
should be treated as a separate risk factor distinct from outdoor air pollution, akin to contaminated
water. In the absence of indoor emissions, indoor concentrations are 30 to 70% lower than outdoors,





especially in colder countries (Chen and Zhao, 2011). This variability in infiltration rates has to be taken
into account for an accurate exposure estimation. Furthermore, it is important to consider how human
exposures can be influenced by outdoor pollution in other environments, such as workplaces and during
commuting, where we spend almost 50% of our times. Health data from citizen cohorts often include
questionnaires that offer valuable insights into the effects of mobility and workplace conditions on
pollution exposure. Overall, while we consider outdoor concentrations at residence place to be a good
proxy of exposure to outdoor pollution, integrating household infiltration rates and mobility data can
significantly help refining exposure estimations.

## 3. Modelling personalized exposures to single PM components

### 3.1 Existing modelling approaches

Figure 2B compares three traditional classes of approaches used for estimating exposures to
PM components. We put forward eight criteria for comparing these approaches including accuracy,
spatial and temporal resolution, spatial and temporal coverage, capability of hindcasting and forecasting
required to estimate past and future exposures and finally, source-specificity and chemical complexity,
i.e. capability to quantify specific PM components. The assessment of the acute health effects requires
the time-series analysis of daily exposures, whereas the link between PM and chronic diseases is based
on long-term exposures determined at high resolution.
Early cohort studies used averaged (Pope Iii et al., 2002) or interpolated (Jerrett et al., 2005)
PM concentrations measured at a few routine monitoring stations to characterize the exposure of
individual participants in different cities. The use of top-down, receptor models based on the
measurements of PM chemical composition has allowed the investigation of PM sources (Belis et al.,
2015; Belis et al., 2020) and their subsequent relation to specific health effects (Ostro et al., 2011).
However, stationary PM measurements are spatially sparse and do not account for the heterogeneity in
pollutant concentrations within cities, especially for primary combustion emissions (Eeftens et al.,
2012b; Elser et al., 2016; Elser et al., 2018). Therefore, several geo-statistical and process-based
chemical transport models (CTMs) have been proposed to fill spatial gaps in long-term descriptions of
PM concentrations.
Land-use regression (LUR) models combine monitoring data with GIS based data, e.g. land
use, traffic, or population density, as emission indicators to predict ground level PM concentrations on
fine grids using regression techniques (Cattani et al., 2017; De Hoogh et al., 2016; De Hoogh et al.,
2013; Eeftens et al., 2016; Hoek et al., 2011; Kim et al., 2016; Wolf et al., 2017). These techniques are
covered in a recent review by (Hoek, 2017). While these techniques are especially pertinent for
modelling primary PM components, e.g. metals (Kim et al., 2016; Chen et al., 2020) or combustion
products (Jedynska et al., 2014; Jedynska et al., 2015), they fail in capturing the overwhelming majority
of PM mass, formed through secondary processes over extended temporal and spatial scales. Therefore,
besides their limited time-resolution (Kim et al., 2016), they have low explanatory power for several
PM components (De Hoogh et al., 2013).
With advances in satellite remote sensing, aerosol optical depth, AOD, measurements of entire
atmospheric columns have been introduced for assessing individual exposure to ground level total PM
mass with much higher accuracy and relatively high time-resolution. Because AOD-PM relationships
are non-linear, interactive and spatiotemporally variable, AOD measurements are typically combined
with other predictors including land-use data and meteorological variables. Models using geo-statistical
and machine learning techniques have been successfully applied at different scales, including city,



regional, national and continental scales as well as in different areas around the world, including EU,
US, and China (Brokamp et al., 2017; Suleiman et al., 2016; Huang et al., 2018; De Hoogh et al., 2018;
Di et al., 2016; Hu et al., 2017; Paciorek et al., 2008; Strawa et al., 2013; Zhan et al., 2017; Di et al.,
2019; Xue et al., 2019; Chen et al., 2018b). However, because they are based on past AOD
measurements, these models cannot forecast future PM concentrations, e.g. as a response to specific
mitigation strategies (Figure 2B). More importantly, they are typically not capable of discriminating
between specific PM components, because AOD measurements of PM columns are not yet chemically
resolved, although future satellite-based sensors will partially deliver this capability (David et al., 2018).
Unlike the other methods, CTMs possess the ability to generate spatial and temporal
distributions of chemically resolved PM components and forecast their future evolutions over large
spatial scales. CTMs are bottom-up, process-based, numerical models, which simulate PM primary
emissions and secondary formation, along with their losses and atmospheric transport in large 3-D
Eulerian gridded domains. Despite their spatial coverage, source-specificity and capability to leverage
complex atmospheric oxidation processes, most CTMs are not sufficiently spatially resolved to be
suited for human exposure assessments (Figure 2B). Due to computational constraints, highly resolved
CTMs are currently limited to city scales, although the application of quantum computing in geoscience
has the potential to overcome these restrictions (Sahimi and Tahmasebi, 2022). As a result, until very
recently, CTM outputs have rarely been exploited for epidemiological analysis, except for optimizing
the retrieval of total PM mass concentrations in AOD-based hybrid models (Di et al., 2019; Xue et al.,
2019) or as an input variable in LUR models (De Hoogh et al., 2016; Shen et al., 2022).
The two fields of air quality modelling, specifically using CTMs and LUR, have evolved along
separate trajectories over the past three decades. This separation can be attributed, in part, to the modest
accuracy of CTMs thirty years ago and, in part, to the substantial contribution of local pollution, such
as traffic, which LUR models were capable of effectively capturing. At that time, CTMs have primarily
focused on implementing representative emission and chemical schemes, aiming to enhance their
accuracy. However, with the advancement in CTMs and the increasing regional nature of PM pollution,
it is now the time for these two fields to converge in order to achieve accurate estimation of exposure
to various PM components at high temporal and spatial resolution and coverage, fulfilling all the criteria
described in Figure 2B.

**3.2 Future directions in fine-resolution modelling of PM components**

More recent modelling developments have allowed the production of fine-resolution maps of
PM chemical constituents on continental (Van Donkelaar et al., 2019; Chen et al., 2020) and global
(Mcduffie et al., 2021; Weagle et al., 2018) scales, including the concentrations of secondary inorganic
aerosols, black carbon, organic aerosols, and dust. These maps were created using a combination of
AOD data and in-situ PM chemical composition measurements to constrain and downscale coarse CTM
outputs to spatial scales commensurate with population density distributions. The resulting maps
offered the possibility to assess the contributions of different anthropogenic emission sectors to regional
and global mortality burden (Mcduffie et al., 2021; Chen et al., 2021b), and to identify which PM
constituents are for example associated with an increased risk of dementia and Alzheimer's disease
(Shi et al., 2023).
These recent developments are a fundamental first step for comprehending the health effects of
individual PM components, but there are limitations to the current approach. Models are still directly
reliant on AOD and in-situ measurements and as such they cannot forecast future PM concentrations
and composition in response to mitigation strategies, global warming, and changes in land-use and



urban build. Additionally, they are limited in identifying the sources of the organic fraction of PM. To address limitations the atmospheric science community should develop hybrid models that instead incorporate land-use data with CTM outputs, enabling the retention of CTMs' source-specificity and forecasting capabilities, while simultaneously benefiting from the fine-resolution information provided by land-use data. In these models, AOD and in-situ measurements should be utilized for model training, rather than as model inputs. CTM-based models have the added benefit of being able to quantify the sources of different constituents, which is especially valuable for the organic fraction, where composition and health effects are heavily dependent on emission sources and formation pathways. To ensure the generation of accurate exposure maps for epidemiological inputs, it is also crucial that exposure models establish connections between air pollution maps and human activity maps and integrate information regarding household infiltration rates. Overall, the development of hybrid models that leverage the complementary strengths of CTMs and land-use information will be key in determining the adverse health effects of different PM components.

## 4. Field observations of PM chemical composition

This section focuses on the type of field observations required to quantify the spatial distributions and temporal variation of PM components and to identify their health impacts.

### 4.1 Established monitoring networks of detailed PM chemical composition

Monitoring networks play a vital role in providing essential data for understanding the spatial distribution and long-term trends of air pollution, identifying emission sources, constraining human exposure models and evaluating the effectiveness of emission reduction measures. International monitoring programs such as SPARTAN[2], EMEP[3], IMPROVE[4], ACTRIS[5] and ASCENT[6] have been critical in establishing and maintaining the operation of these networks. Besides the continuous provision of detailed PM measurements for policymaking, these monitoring programs offer access to outstanding facilities and openly available databases for scientists from academia and the private sector, promoting cutting-edge science and international collaborations.

Another advantage of these programs is the standardization of analytical approaches and data formats, which ensures data quality and comparability and facilitate data sharing and use. Data generated from these programs may include particle number-size distributions and the concentrations of elemental and organic carbon, major ions and metal components. Figure 3 illustrates the distribution of stations across Europe where we have gathered detailed PM chemical properties generated from different national and pan-European programs. For some PM constituents, more than 50,000 daily

---

[2]SPARTAN: Surface Particulate Matter Network (SPARTAN) provides publicly available data on PM mass, chemical composition, and optical characteristics for connection with satellite remote sensing and for air quality management.
[3]EMEP: European Monitoring and Evaluation Programme aims to monitor and model the long-range transport of air pollutants across Europe.
[4]IMPROVE: Interagency Monitoring of Protected Visual Environments is a long-term monitoring program designed to assess the visibility and air quality in national parks and wilderness areas in the United States. The primary goal of the IMPROVE network is to measure PM mass and chemical composition, at over 170 monitoring sites across the United States.
[5]ACTRIS: Aerosol, Clouds, and Trace gases Research InfraStructure is a pan-European research infrastructure of several measurement stations across Europe that provides long-term observational data on aerosols, clouds, and trace gases.
[6]ASCENT: The Atmospheric Science and Chemistry mEasurement NeTwork is a new comprehensive, high-time-resolution, long-term measurement network in the U.S. for the characterization of aerosol chemical composition and physical properties.





concentrations at different sites are available, which is rare, if not unique. This is only possible thanks
to such research infrastructures. Datasets of at least this scale are required to form a complete picture
of the PM chemical and physical properties and sources, with which our atmospheric modelling
community can optimize exposure maps to understand the health effects of different PM constituents
on a continental level.

The composition, emission sources and formation pathways of the organic fraction remain a
scientific challenge. Routine measurements (e.g. of organic carbon) are not sufficiently chemically
resolved for the retrieval of the contributing sources. For this, two approaches are currently exploited
for long-term monitoring: the aerosol chemical speciation monitor, ACSM (Ng et al., 2011; Fröhlich et
al., 2013), which measures the bulk composition of the non-refractory fraction of fine PM and infrared
spectroscopy, IR (Weakley et al., 2016), which measures the functional group composition of the
organic fraction. We have utilized ACSM data to determine the contribution of residential emissions,
vehicular emissions and secondary processes to the organic aerosol fraction across Europe (Chen et al.,
2022) and to validate CTM outputs (Ciarelli et al., 2017; Jiang et al., 2019). ACSM measurements are
part of ACTRIS and ASCENT, whereas the IMPROVE network has adopted IR measurements. The
complex composition of the organic aerosol, especially of the oxygenated secondary fraction, means
that no technique is complete. The spectra acquired with both ACSM and IR techniques retain
information on the source origins and the formation pathways of the organic fraction. These two
techniques are complementary and their combination, although currently only exploited in the
laboratory (Yazdani et al., 2021, 2022), can be very powerful to further characterize the organic aerosol
fraction in dense networks over long-terms, enabling a better understanding of the relationship between
its composition and health effects.

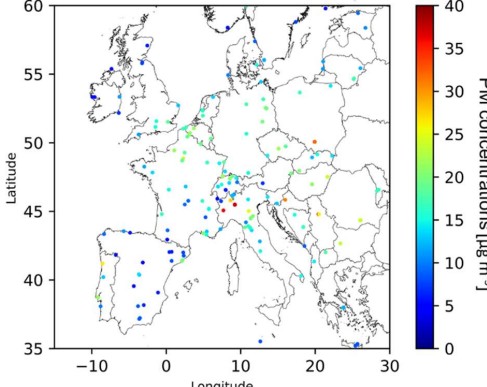

**Figure 3:** European map with site locations where long-term detailed chemical composition data is
available. Sites are both urban and rural. Markers are colour-coded with total annual PM concentrations
in 2013, to reflect differences in emissions between sites.

Overall, it is essential that the scientific community continues to leverage chemically-speciated
PM data from monitoring networks and generates additional datasets for validating exposure models. It
is also vitally important that governments continue investing in these networks to foster innovative
research in the field of air quality and health.





### 4.2 Why detailed atmospheric chemistry matters – a comparison of severe PM pollution in Northern China and Northern India

Modest improvements in PM pollution in relatively clean regions in Western Europe and North America, where most of the current monitoring programs operate, would result in large avoided mortality, owing to the nonlinear concentration-response relationships that describe the risk of death against PM exposures (Apte et al., 2015). At these locations, air quality is very sensitive to the contribution of natural emissions, which means further air quality improvements are more subject to the whims of nature (Figure 4). For these locations, it is crucial to intensify efforts to quantify natural emissions and collaborate closely with the WHO to identify effective strategies to exempt them from guidelines.

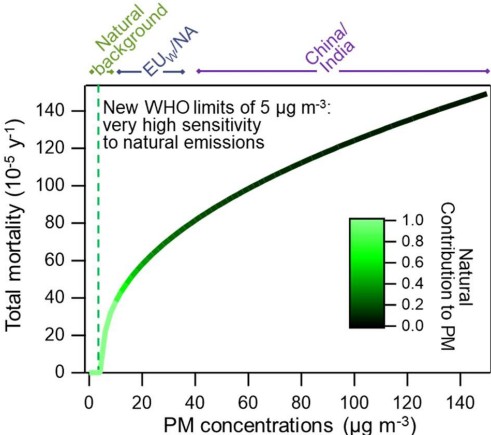

**Figure 4:** Dose-response relationship between PM concentrations and total attributable mortality, highlighting the sensitivity of mortality to reductions in anthropogenic emissions at low and high pollution levels and potentially to contributions from natural emissions – adapted based on (Apte et al., 2015). Vertical axes indicate per-capita mortality rates attributable to $PM_{2.5}$ for a hypothetical global population uniformly exposed to a given level of $PM_{2.5}$. The dose-response relationship is coloured by the contribution of natural emissions to PM mass. The horizontal bars at the top of the figure represent typical PM concentrations in Western Europe/north America ($EU_W$/NA) and China/India, as well as natural background PM concentrations.

By contrast, major improvements in air quality would be required to substantially reduce mortality in more polluted regions, such as China and India (Figure 4), although such improvements are at least possible as high concentrations result from anthropogenic activities, and are therefore more controllable. Air pollution in China and India together causes approximately 5 million deaths every year (Lelieveld et al., 2015), with approximately 20% of the total deaths attributable to PM (Figure 5A). Projected demographic shifts in these regions indicate that in order to maintain current PM-attributable mortality rates, average PM levels must decrease by approximately 30% within the next 15 years to counterbalance the rise in PM-related deaths resulting from aging populations (Apte et al., 2015). Therefore, an effective program to deliver clean air to polluted regions is urgently needed to avoid several million premature deaths every year.

In response, China and India launched their country-level clean air plans in 2013 and 2019, respectively. Despite greatly improved national air quality levels compared to ten years ago (Figure 5B), China is now finding further air pollution reduction challenging due to the trade-off between controlling PM and ozone pollution (Li et al., 2019). The situation in India is more alarming. The



country's air quality continues to worsen despite the implementation of its clean air program. A growing number of cities experience severe pollution (Ghildiyal, 2022), resulting in a rise of the mortality attributable to PM pollution (Figure 5B). The mechanism of haze formation in the two regions is also very different. While pollution in China happen on regional scales, local pollution in India plays a prevailing role. The comparison between severe PM pollution in Northern China and Northern India serves as a perfect example for why a detailed understanding of the complex atmospheric chemistry involved is required to mitigate the air pollution problems and health effects in those regions.

In China, secondary aerosol production was identified as the main cause behind winter haze events in a study conducted by Huang et al. (2014), which was the first of its kind to make this discovery a decade ago (Huang et al., 2014). Later studies have confirmed that in Chinese megacities, particle formation, often observed at the onset of haze, is driven by the photochemical production of secondary organic and inorganic species, which happens on a regional scale during the day (Yao et al., 2018; Kulmala et al., 2021). The high concentrations of anthropogenic sulfate and nitrate, coupled with high relative humidity, provide an additional reactive medium for heterogeneous aerosol production (Tong et al., 2021), further contributing to haze formation (Le et al., 2020). Because of the nonlinear chemistry of ozone production and titration in winter, the recent reductions in nitrogen oxides result in ozone enhancement in urban areas (Li et al., 2019), further increasing the atmospheric oxidation capacity and facilitating secondary aerosol formation (Le et al., 2020). Substantial oxidation in China's atmosphere is at play even during the night. New findings reveal that between 2014 and 2019, the decrease in pollution has led to an increase in the production rates of nitrate radicals across China, suggesting the growing role of nighttime chemistry to China's air pollution (Wang et al., 2023a). Further mitigating air pollution and its health effects in China will require a detailed understanding of the complex atmospheric chemistry behind oxidant production, as well as the identification of the major sources of secondary aerosol precursors.

In Delhi-India, however, the rapid growth of particles into sizes relevant for haze formation occurs during nights without any photochemistry. We have recently shown that the growth of sub-100 nm particles is predominantly driven by primary supersaturated organic vapors from local biomass combustion emissions, whose condensation is promoted by the rapid decrease in air temperature and the increase in emissions during nighttime (Mishra et al., 2023). The formation of ammonium chloride enhances aerosol water uptake through co-condensation at high nighttime relative humidity, which sustains particle growth at higher sizes (Mishra et al., 2023) and leads to fog formation and a 50% reduction in visibility (Gunthe et al., 2021). This process, apparently unique to India's capital, does not involve photochemistry but is instead driven by high emissions of hydrochloric acid, possibly from local industries (Rai et al., 2020). During daylight hours, with the dispersion of $NO_X$ emissions and the increase in the atmospheric oxidation capacity, local combustion of fossil fuels and biomass become an important source for SOA production (Kumar et al., 2022). Toxic heavy metal pollution levels in Delhi are another cause for alarm, with concentrations several hundred times higher than those found in Europe, also due to local industries (Rai et al., 2021). Solving air pollution in India will require international collaboration with local researchers to better understand the local sources of different pollutants, e.g. through mobile measurements (Section 4.3), as well as the effects of local meteorological conditions on air quality. Given the significance of local pollution sources, it will also necessitate the involvement of social scientists and local communities to introduce social changes and raise public awareness.



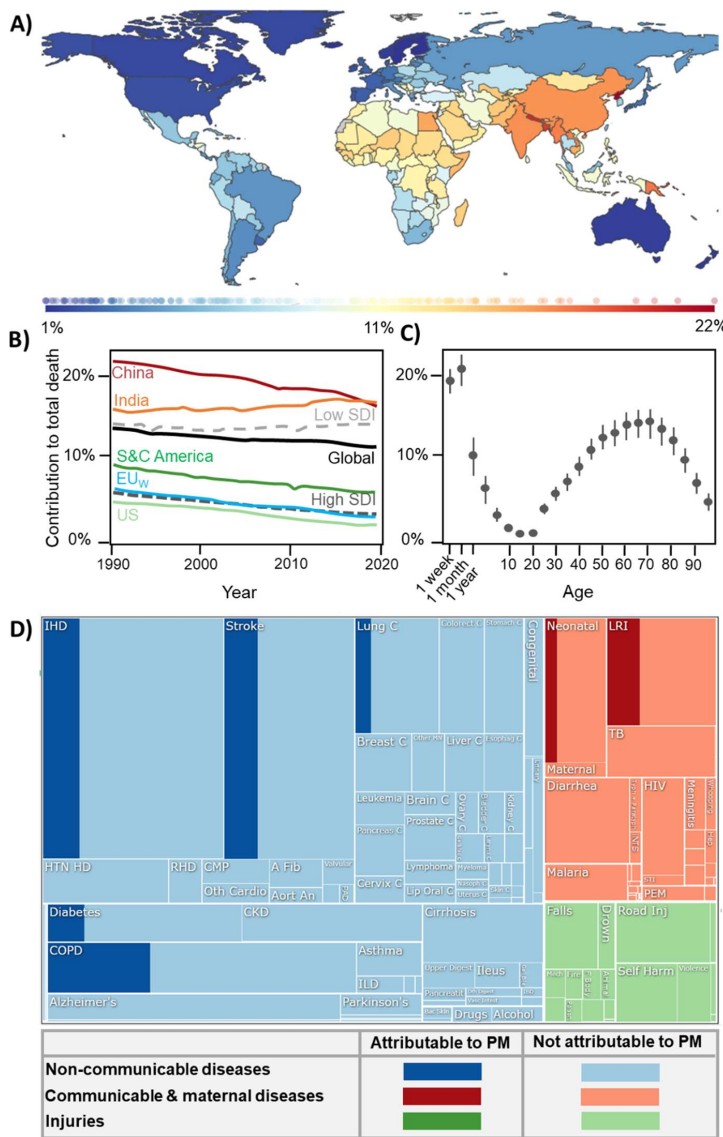

**Figure 5: percentage of mortality attributed to particulate pollution.** Data are from the Global Burden of Disease Study 2019 Results (Seattle, United States: Institute for Health Metrics and Evaluation, 2020 - available from https://vizhub.healthdata.org/gbd-results/). A) Percentage of PM-related mortality for every country. B) Evolution of the percentage of PM-related mortality from 1990 to 2019 for locations discussed in the text, including China, India, Western Europe (EU$_W$), US, South and Central America, low SDI (Socio-demographic Index) and high SDI. C) Percentage of PM-related mortality globally as a function of age. D) Percentage of deaths attributable to PM pollution related to non-communicable diseases, communicable & maternal diseases and injuries. The main causes of death to which PM exposure contribute include ischemic heart diseases (IHD), stroke, diabetes, chronic obstructive pulmonary diseases (COPD), neonatal infections, and lower respiratory infections (LRI).

The atmospheric science community has already made significant strides in understanding the sources of air pollution in China and India, but knowledge gaps still exist. It is imperative to further understand the non-linear effects of emissions on the atmospheric oxidation capacity, particularly in



light of India's potential to face the same problems as China in the near future when primary pollution
reduction will lead to an increase in the photochemical production of ozone and secondary aerosols. It
is also crucial to identify on a molecular level the specific ingredients contributing to aerosol formation
and growth and relate these ingredients to the emission sources of their precursors. We also need to gain
a mechanistic understanding of the interplay between the soluble inorganic fraction and water and their
effects on the enhanced partitioning and heterogeneous chemistry of organic and inorganic vapors (e.g.
$N_2O_5$, HCl, $HNO_3$, and oxidized organics). Without this knowledge, we cannot accurately predict the
fate of these vapors with future reductions in the anthropogenic emissions of inorganic precursors, such
as $SO_2$ and $NO_X$.
Finally, it is essential to establish national monitoring networks in both countries that probe the
spatial distribution and long-term trends of air pollution, and allow us to evaluate the effectiveness of
emission reduction measures. The data resulting from these monitoring programs serve as a cornerstone
for understanding the health effects of the PM components specific to China and India, enabling us to
devise regionally-specific solutions aimed at effectively limiting air pollution in these regions. More
generally, the inequity of air pollution is flagrant, with locations having low socio demographic index
(SDI) suffering three times the burden of PM-related mortality compared to locations with high SDI
(Figure 5B). This disparity underscores the urgent need for comprehensive monitoring networks in low
SDI countries, enabling proactive measures to mitigate the health impacts of PM pollution.
**4.3 Fine-resolution measurements of urban pollution**
Monitoring networks have limited spatial coverage, which can make it difficult to capture
localized pollution hotspots, especially from primary combustion emissions (Eeftens et al., 2012b; Elser
et al., 2016; Elser et al., 2018; Jedynska et al., 2015; Jedynska et al., 2014). Therefore, several
approaches have been proposed for the spatial measurements of urban pollution (Figure 2). Both
ground-based sensor networks, e.g. for $CO_2$, black carbon, $NO_2$, or total PM (Popoola et al., 2018;
Caubel et al., 2019; Oney et al., 2015), and satellite retrievals (Di et al., 2016; Griffin et al., 2019) can
map the concentrations of individual pollutants at sub-km-scale resolutions, however, these approaches
lack the chemical resolution needed for the measurements of PM components. Aircraft measurements
are suited for studying pollution plumes at regional scales (Fry et al., 2018; Decker et al., 2019), but
cannot access fine scale variations at the ground level. Ground-based mobile laboratories can house
online instrumentations that provide high chemical resolution, while operating with sufficiently high
time resolution (i.e. few minutes) for measurements at street levels (Shairsingh et al., 2018; Gu et al.,
2018). This makes them ideally suited for spatial mapping of specific atmospheric pollutants in urban
environments and for model verifications (Hankey and Marshall, 2015; Alexeeff et al., 2018; Apte et
al., 2017; Gu et al., 2018).
A large number of studies have measured black carbon, $NO_2$, total PM mass and number
concentrations aboard of mobile platforms (Alexeeff et al., 2018; Apte et al., 2017; Hankey and
Marshall, 2015; Shairsingh et al., 2018; Simon et al., 2017; Miller et al., 2020). The Aerodyne aerosol
mass spectrometer (AMS) has also been used with a great effect for the mobile measurements of non-
refractory PM components, including secondary inorganic species and organic aerosol (Elser et al.,
2016; Elser et al., 2018; Gu et al., 2018; Mohr et al., 2011; Shah et al., 2018). The application of
factorization techniques to the measured organic mass spectra has even enabled its apportionment to
primary traffic, cooking and biomass burning emissions as well as the quantification of a total secondary
fraction (Gu et al., 2018; Elser et al., 2016; Elser et al., 2018). From measurements in the EU and the
US, it was found that the secondary organic and inorganic fractions are homogeneously distributed

https://doi.org/10.5194/egusphere-2023-1472



across cities, while primary emissions are enhanced by several µg m$^{-3}$ compared to background levels
(Elser et al., 2016; Elser et al., 2018) in correlation with land-use variables (Gu et al., 2018).
Until recently, there has been no robust technology for highly time resolved measurements of
airborne particulate metals. Therefore, studies had previously relied on integrated offline samples
collected over days-to-weeks at only a few sampling stations in order to assess the spatial distribution
of particulate metals across cities (Li et al., 2016; Van Ryswyk et al., 2017; Zhang et al., 2015). With
such measurements, intra-urban variability in metal concentrations can still be discerned. However, due
to the limited sample sizes (less than five samples per site and 200 samples in total) and the low time
resolution of sampling, robust land-use regression models of daily exposures to toxic metal particles
cannot be achieved. Recently, the Xact 625 ambient metals monitor, an online XRF spectrometer, has
been developed and successfully deployed in the field for the real time measurements of particulate
elements (~25)  with time resolutions down to 30 minutes (Furger et al., 2017). Due to its high time
resolution, sensitivity and robustness in the field, the Xact is capable of delivering several month long
datasets of 1000s of data points – 10-100 times more than offline techniques (Manousakas et al., 2022),
which allow the retrieval of daily exposure patterns. However, further developments are needed to
achieve particulate elemental analysis on time-scales of minutes suitable for mobile measurements. The
availability of such measurements will enable access to the aerosol's elemental composition at a fine
resolution, which is necessary for validating exposure models for metal components.
Street-level air quality data can enhance, challenge, or confirm various air quality datasets, such
as regulatory data, CTM outputs, land-use regression predictions, and remotely sensed observations.
This refinement can aid addressing exposure misclassifications in epidemiological studies (Zeger et al.,
534    2000).

## 5. Gaps in understanding emissions and their health impacts

Human activities have significantly altered the earth's environment, leading to profound
changes in the atmospheric composition, global temperatures and land cover. In Figure 6, we categorize
the complex anthropogenic effects on PM concentrations and composition into four broad classes:
**(1) Direct emissions:** This class includes anthropogenic pollutants that are released directly into the
atmosphere.
**(2) Land-use changes:** Alterations in land use have a direct impact on PM levels and composition.
These changes encompass modifications in build environments, urban greening initiatives,
deforestation/forest management, and agricultural practices. These changes affect emissions, pollutant
accumulation, and exposure patterns, such as the "street canyon effect". Understanding these influences
is crucial for an accurate quantification of personalized exposure to PM components.
**(3) Direct effects of anthropogenic emissions on the chemistry of natural PM.**
**(4) Indirect perturbation of natural PM:** anthropogenic emissions can indirectly influence natural
PM through their impacts on natural ecosystems, e.g. through global warming, increased $CO_2$
concentrations, shifts in vegetation patterns, or desertification.
With this section, we address existing gaps in our understanding of anthropogenic emissions,
their atmospheric transformation processes, and their direct and indirect influence on natural PM. It is
important that the atmospheric science community approach these gaps from a mechanistic standpoint
and incorporate them into models to accurately quantify the anthropogenic impacts on PM components
and their associated health effects. This distinction between controllable and uncontrollable emission



sources, as well as detrimental and non-detrimental ones, serves as a key first step in developing targeted
mitigation strategies. In section 5.1, we delve into anthropogenic PM emissions that hold particular
relevance for public health, while in section 5.2, we focus on the direct and indirect effects of
anthropogenic emissions on natural PM.

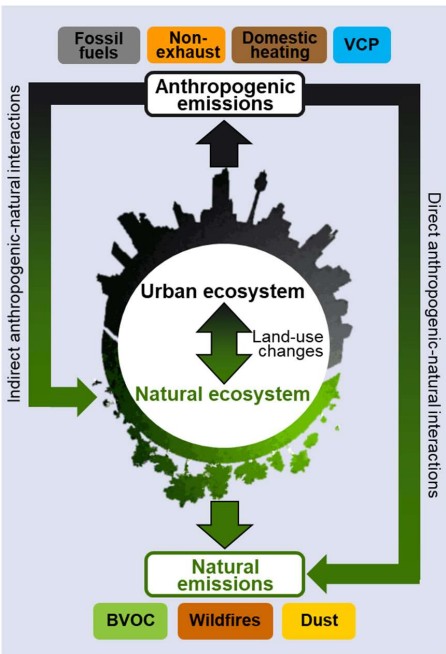


**Figure 6: anthropogenic effects on PM through (1) direct emissions, (2) land-use changes, (3) direct and (4)**
**indirect perturbation of natural PM**. (3) comprises the direct effects of anthropogenic emissions on the
chemistry of natural aerosols, while (4) describes the influence of anthropogenic emissions on natural ecosystems,
e.g. through global warming, or increase in $CO_2$ concentrations. Natural emissions from terrestrial systems include
biogenic volatile organic compounds (BVOCs), wildfire emissions and dust. Anthropogenic emissions include
$NO_X$ and $SO_2$ from fossil fuel combustion, non-exhaust emissions, solid fuel combustion for domestic heating,
and volatile chemical products (VCPs).

## 5.1 Health effects of anthropogenic PM emissions

Anthropogenic emissions remain a predominant source of primary and secondary aerosols. Our
review reveals mixed results regarding the differential health effects associated with different
anthropogenic PM components (Chen et al., 2018a; Yang et al., 2019; Masselot et al., 2022; Wang et
al., 2022), but with elemental carbon, organic aerosols and sulfate consistently associated with
increasing risks of mortality and hospitalization. We believe that one crucial factor contributing to the
inconsistencies in these findings is the strong correlation between various PM components. For
example, while sulfate itself may not be toxic, it often exhibits a strong correlation with SOA, provides
a medium for organic reactions, and influences the bioavailability of dust elements. Therefore, a holistic
consideration of the correlations among PM components is essential when analysing their differential
toxicities. There is a pressing need for extended datasets that provide high spatial and temporal coverage
and resolution, allowing overcoming limitations related to the covariance between PM components.
Additionally, it is crucial to achieve a detailed separation and characterization of PM components, with
a special focus on the OA. In this section, we will focus on emissions that will become increasingly



important for public health in the future, including non-exhaust on-road emissions, volatile chemical products (VCPs), and residential biomass burning.

As traffic exhaust emissions of $NO_X$, PM and hydrocarbon vapours are increasingly regulated, car engines have undergone a technological revolution, improving combustion efficiency and after-treatment technologies. In contrast, non-exhaust emissions, such as brake and tire wear, have increased with the growing number of vehicles and currently exceed exhaust emissions (Timmers and Achten, 2016). These emissions control toxic metals such as copper, which enhance the oxidative potential of PM (Daellenbach et al., 2020). Even with the electrification of the fleet, non-exhaust emissions will remain an issue, potentially worsened by the heavier weight of electric cars (Timmers and Achten, 2016). While public transportations, including trams and trains, may also be an important source of metal particles, their contribution are not yet well quantified. Atmospheric scientists must comprehend the distribution of on-road non-exhaust emissions to quantify their health impacts.

With the drastic reduction of on-road transportation emissions, VCPs have emerged as the largest source of urban organic emissions in US and European cities, modulating urban chemistry (Coggon et al., 2021; Gkatzelis et al., 2021; Mcdonald et al., 2018). These ubiquitous emissions encompass pesticides, coatings, printing inks, adhesives, cleaning agents, and personal care products. Human exposure to fossil carbonaceous aerosols and to ozone is transitioning from transportation-related sources to VCPs. These emissions have comparable, if not greater, SOA potentials compared to vehicular emissions, which may influence human health. Variations in SOA potentials and chemistry among VCPs, as revealed by laboratory experiments, highlight the need for further characterization of these unconventional emissions (Shah et al., 2020). Furthermore, it is now possible to include these emissions into models (Pennington et al., 2021), which will enable future assessments of their health impacts. Existing regulations on VCPs emphasize reducing ozone and air toxics, but currently exempt numerous chemicals that contribute to SOA formation. Efforts to refocus mitigation strategies for ozone formation and toxic chemical burdens require atmospheric scientists to provide data quantifying the contribution of these emissions to the global burden of disease.

Achieving net-zero emissions for climate goals does not necessarily guarantee clean emissions for air quality. Biomass combustion, adopted as a carbon neutral energy source for residential heating, is a potent anthropogenic source of pollution during winter. It dominates the emissions of toxic organic species such as polycyclic organic compounds. The emitted organic vapours rapidly react in the atmosphere with OH and $NO_3$ radicals, resulting in substantial SOA production (Kodros et al., 2020; Stefenelli et al., 2019). The SOA formed contains high levels of oxygenated and nitro-aromatic compounds, which likely cause the high oxidative potential of this fraction (Daellenbach et al., 2020). Recent laboratory investigations (Liu-Kang et al., 2022; Wang et al., 2023b) and airborne field measurements (Morgan et al., 2020; Zhou et al., 2017) suggest that primary biomass emissions, which absorb near UV light, can undergo photoreactions in the particle phase, resulting in a doubling of the emissions oxidation state in few hours. The dominant transformation processes of biomass burning emissions and their impact on aerosol toxicity remain unclear. Overall, biomass smoke has not shown a reduction trend in many regions worldwide, underscoring the importance of comprehending the fate of these emissions in the atmosphere and their implications for human health.

## 5.2 Anthropogenic effects on natural PM and implications for health outcomes

With the increasing regulations on anthropogenic emissions, the contribution of natural emissions, including biogenic volatile organic compounds (BVOCs), wildfires and desert dust, will gain prominence (Figure 4). While these emissions stem from natural ecosystems (Figure 6), they are also



significantly perturbed by anthropogenic emissions. The traditional picture that distinguishes biogenic
and anthropogenic emissions obscures human impacts on ostensibly natural systems. Anthropogenic
effects on natural PM can be either direct, through the alteration of atmospheric reactivity, or indirect,
through feedback mechanisms triggered by changes to the biosphere. In this section, we discuss changes
in natural PM emissions that are important to consider when determining their impacts on human health
(Table 1).
Biogenic SOA (BSOA) is the most important source of OA in the atmosphere (Jiang et al.,
2019), with mobile sources of $NO_X$ playing a vital role in moderating its formation, composition and
potentially health effects. $NO_X$ effects on BSOA are multifaceted and involves (1) altering the fate of
biogenic $RO_2$ radicals, (2) increasing the atmospheric oxidant concentrations and (3) providing an
aqueous medium for additional reactions (Xu et al., 2015; Pye et al., 2019; Carlton et al., 2018). As
NOx emissions decrease, $RO_2$ autoxidation becomes increasingly important, potentially enhancing
BSOA formation, while oxidant availability driving $RO_2$ formation rates simultaneously declines,
possibly slowing regional BSOA formation. Recent modelling analyses (Carlton et al., 2018), along
with in-situ (Xu et al., 2015) and airborne measurements (Pye et al., 2019; Shrivastava et al., 2019)
consistently suggest that anthropogenic $NO_X$ leads to a net enhancement in BSOA concentrations by
20-50% depending on the location and season. Similar to $NO_X$, $SO_2$ emissions from electricity
generation, the main source of particulate sulfate, modulates the aqueous formation of isoprene SOA.
Models (Carlton et al., 2018) and measurements (Xu et al., 2015) over the US demonstrate that between
40–70% of the BSOA can be controllable by reducing anthropogenic $NO_X$ and $SO_2$. Similar analysis is
still lacking at other locations worldwide.
BSOA concentration exhibits a strong temperature dependence, driven by the exponential
increase in BVOC emissions and their oxidation rates. Our analysis of multi-field observational datasets
from European and North American locations reveals that BSOA contributes [0.9–2.5] $\mu g\, m^{-3}$ at 15°C,
compared to [2.1–6.3] $\mu g\, m^{-3}$ at 25°C (Xu et al., 2015; Daellenbach et al., 2017). Climate models project
a global increase in BSOA mass by approximately 30–150% with a temperature rise of 2°C and a few
hundred ppb increase in atmospheric $CO_2$ concentrations (Carslaw et al., 2010). When changes in
vegetation are accounted for, predictions of BVOC emissions become extremely uncertain, with
projected increases ranging from 10s to 100s of percent. These uncertainties arise from the
unpredictable response of vegetation to future climates, including longer growing seasons, increased
leaf area index with the fertilization effect of $CO_2$, changes in water stress and expansion of the boreal
and temperate forests. With the rise in BVOC emissions and the denitrification of the atmosphere, it is
expected that the oxidation capacity of the atmosphere may decrease leading to slower production of
BSOA and a complete change in its composition. Understanding the non-linear interactions among
anthropogenic emissions of oxidant precursors, greenhouse gases, atmospheric oxidation conditions,
and the biosphere is crucial for understanding BSOA concentration and chemical composition.
While reducing $NO_X$ and $SO_2$ can control a significant portion of BSOA, the rise in BVOC
emissions with climate change, albeit highly uncertain, may offset this reduction. Currently, there is
very limited understanding of the impact of BSOA on human health, with only one study suggesting a
3.5 times higher cardiorespiratory mortality associated with anthropogenically-influenced BSOA
compared to total PM (Pye et al., 2021). This complex interplay between anthropogenic emissions,
BSOA production, chemical composition, and their impact on human health remains highly uncertain.
Atmospheric scientists should capitalize on emerging multi-year, multi-location observations of
detailed PM chemistry to enhance model predictions of BSOA chemical composition, burden and



response to global changes and estimate the effect of this fraction on different health outcomes for
different regions worldwide (Table 1).

**Table 1:** Future changes in natural emissions, key observations needed for coupling with health data, high priority model developments for understanding the health effects of emissions and their future evolution, and level of scientific understanding (LOSU) of natural PM health effects

| Source | Future changes | Key observations | Model developments | LOSU |
|---|---|---|---|---|
| BSOA | Increase in global BSOA burden by 30-150%. | Long-term, multi-site measurements of BSOA precursors, oxidant precursors, chemistry and burden. <br><br> Global analysis of response of BSOA chemistry and burden to anthropogenic emissions and climate change. <br><br> Fundamental studies of anthropogenic-biogenic interactions and their effects on BSOA chemistry and burden | Improving the understanding of the response of BVOC emitting species to climate change (temperatures, soil nutriments, $CO_2$, nitrogen deposition, droughts, vegetation shifts). <br><br> Implementing the effects of anthropogenic-biogenic interactions on BSOA chemistry and burden | Very poor understanding of the effects of BSOA on chronic and acute health outcomes. <br><br> Very poor understanding of the impact of anthropogenic emissions on BSOA health effects. |
| Wildfires | Increase in wildfires frequency by ~100% and emission burden by ~30%. | Long-term global records of fire occurrence and associated PM emissions. <br><br> Global analysis of response of wildfire emission occurrence and budget as function of climate change, and fire drivers (temperature, droughts, lightning). <br><br> Determination of wildfire emission rates for different ecosystems. <br><br> Fundamental studies of wildfire emission and their atmospheric transformation processes. | Coupling fire and vegetation models. <br><br> Improving the understanding of the impact of land and fire management on fire emissions. | Medium understanding of the effects of wildfire emissions on acute health outcomes, mainly related to respiratory complications. <br><br> Poor understanding of the effects of wildfire emissions on chronic health outcomes, mainly related to different cancer sites. |
| Dust | Uncertain | Long-term global records of dust emission burden, size and chemical composition. <br><br> Quantification of the contribution of soil vs. urban dust in major cities. <br><br> Field and laboratory observations of dust aging and its impact on the bioavailability of key elements. | Improving dust emission schemes. <br><br> Implementation of dust updated aging schemes. | Medium understanding of the effects of dust emissions on acute health outcomes. <br><br> Poor understanding of the effect of dust origin and aging on its health effects. |

Wildfires have become increasingly frequent in many regions worldwide, making them the
second largest contributor to atmospheric organic carbon on a global scale. This source can be directly
affected by anthropogenic activities, through deforestation, forest management and fire suppression or
indirectly by climate change. Climate models predict that global warming will amplify wildfire
emissions by ~30%, owing to longer fire seasons, higher temperatures, increased droughts, and
increased convection-induced lightning as an ignition source (Carslaw et al., 2010). The short-term
health effects of wildfire emissions, including pulmonary complications (Stawovy and Balakrishnan,
2022), respiratory mortality and cardiovascular mortality (Chen et al., 2021a), have been firmly
established. Conversely, understanding the long-term health effects of these emissions is an ongoing



area of research (Grant and Runkle, 2022; Gao et al., 2023). Recent studies investigating Amazonian
(Yu et al., 2022) and Canadian Boreal (Korsiak et al., 2022) wildfire emissions have highlighted an
elevated risk of various cancers, surpassing the effects of non-wildfire PM emissions for equivalent
exposure doses. With the projected increase in wildfires, it is imperative for atmospheric scientists to
comprehensively comprehend wildfire emissions, assess their health impacts and predict their future
evolution. A crucial first step in this direction is the analysis of global fire occurrence records and
associated PM emissions. Such analysis will establish robust relationships between emissions,
ecosystems, climate change, fire management and fire drivers. These records will also form the
foundation for improving our understanding of short and long-term health effects of wildfires (Table
689  1).

Dust is the most important source of elements in the atmosphere, affecting public health and
through deposition modulating nutriment availability, the carbon cycle and biogeochemistry in oceanic
and forest ecosystems. Wind speed, soil moisture and vegetation cover are the main drivers of dust
emission fluxes, size distribution and mineralogical composition (Carslaw et al., 2010). During
transport, dust particles react with acids, reducing their lifetime against wet deposition and increasing
the bioavailability of key elements, including Iron. It has been shown that anthropogenic sulfate from
fossil fuel combustion modulates soil dust iron solubility and toxicity (Wong et al., 2020). In addition
to sulfuric acid, nitric acid may be associated with dust particles to a notable extent. The reactive uptake
of gases with dust particles heavily depends on dust mineralogical composition, with particles rich in
carbonates exhibiting strong atmospheric reactivity. Similar to wildfire emissions, both direct human
activities and climate change can influence dust emissions, making future predictions uncertain
(Carslaw et al., 2010). Dust outbreaks have frequently been associated with mortality and hospital
admissions (Stafoggia et al., 2016; Crooks et al., 2016), albeit with moderate effects and associated high
uncertainties in risk rate estimates (Zhang et al., 2016). This uncertainty may result from the variability
in dust particles morphology, size, solubility and chemical composition, depending on their origin and
transport time in the atmosphere. Additional observational data on dust phenomenology is required for
model evaluation. A particular challenge is the provision of long-term, large scale datasets, which is
crucial because of the strong spatial and temporal variability of dust concentration, size and chemistry
in the atmosphere. Finally, fundamental research on dust transformation processes and their impact on
health effects is warranted (Table 1).
In this section, we have discussed the key observations and modelling developments that are in
our opinion needed to represent different anthropogenic and natural emissions and comprehend their
health effects. While anthropogenic emissions are destined to decrease, natural emissions will most
likely increase. Part of this increase can be controllable through reducing anthropogenic emissions and
managing land-use. The atmospheric science community is now ready to provide the field
measurements, laboratory observations and model outputs needed to quantify the contribution of
anthropogenic, controllable and uncontrollable natural emissions globally and predict their evolution
with global changes. Such data will constitute the foundation for a constructive dialogue with
stakeholders and policy makers for finding the best ways for exempting uncontrollable natural
emissions from guidelines.

## 6. Collaboration between atmospheric scientists and epidemiologists

This section highlights the critical role of the collaboration between epidemiologists and
atmospheric scientists in identifying the specific PM components responsible for various diseases and



elucidating the underlying biological pathways through which these components can trigger disease
progression.

### 6.1 A step towards causality with population-based epidemiology

Recently, eight hallmarks of environmental insults have been proposed (Peters et al., 2021).
They encompass oxidative stress and inflammation, genomic mutations, epigenetic alterations,
mitochondrial dysfunction, endocrine disruption, altered intercellular communication, changes in
microbiome communities, and impaired nervous system function. These hallmarks jointly underpin the
severe health effects resulting from lifelong environmental exposures, even to relatively modest
concentrations of contaminants.
Barrier organs, such as the lung or the gut, are directly impacted by environmental exposures
and have evolved to cope with insults. The immune function within these organs serves as the first line
of defense, while our sensory system may elicit neurological responses to adapt to changing
environmental conditions. However, environmental impacts extend beyond immediate and local
responses caused by acute exposures. Recurring local reactions from chronic exposures can trigger
systemic responses beyond the initial site of the insults, activating the immune system, triggering
metabolic functions, altering organ-to-organ signaling, disrupting autonomic nervous system control,
and affecting the genetic expression. These responses are geared at maintaining the homeostasis of
organ functions and, most importantly, determine wellbeing and disease development.
PM, as one of the most important environmental insults, can enter our body through various
barriers, e.g. our lungs or digestive system, affecting individuals through the complex web of biological
pathways mentioned above. Figure 5C displays the contribution of PM pollution to total mortality at
various ages, illustrating the staggering effects of PM for infants and elderly individuals. Short-term
exposure to PM pollution has been linked to sudden infant death and higher mortality and morbidity
rates, caused by cardiorespiratory issues, renal complications, and mental disorders. These effects are
particularly pronounced in children and individuals with chronic conditions (Heft-Neal et al., 2018)
(Zhang et al., 2023; Liu et al., 2023; Guo et al., 2023). According to a recent multi-location assessment,
every 10 µg m$^{-3}$ increase in daily PM levels increases the mortality risk by 0.7% (Liu et al., 2019).
Conversely, long-term PM exposure has been linked to numerous non-communicable diseases that
manifest at a later stage of life (Figure 5C), including cardiovascular diseases (Requia et al., 2017;
Lelieveld et al., 2019), respiratory symptoms (Nhung et al., 2017; Zheng et al., 2015) , different types
of cancers (Turner et al., 2020), diabetes (Yang et al., 2018), and neurodegenerative diseases (Maher et
al., 2016; Shi et al., 2020).
Unlike infectious diseases, non-communicable diseases have multiple causes and involve
various factors, which individually are neither necessary nor sufficient to cause the disease. Early-life
exposures may leave enduring marks in the body, leading to manifestations that can arise many decades
later. Given the multifactorial nature of the problem, epidemiology is irreplaceable when it comes to
investigating non-communicable diseases and working with citizen cohorts is essential to circumvent
the challenges of randomization taking into consideration cofounding effects. Citizen involvement is
simply inevitable in comprehending their own health.
Epidemiologists rely on patterns to infer potential cause and effect relationships, before fully
understanding the underlying biological pathways. The epidemiological associations between PM
exposure and diseases are consistently and unequivocally established. To bolster the causal
interpretation of these associations, it is crucial to identify the intermediate steps that connect exposure



and disease. Therefore, we must focus on developing tools to investigate which of the eight hallmarks are involved in disease development and detect early changes at low PM doses. In this regards, epidemiology may greatly benefit from advancements in environmental characterization, molecular phenotyping, multiomics, epigenetics, imaging, as well as the implementation of personalized and digital medicine (Probst-Hensch et al., 2022). The integration of these tools have the potential to transform modern population-based environmental epidemiology, advancing our understanding of disease etiology and enabling the connection of exposure to the development of specific disease hallmarks. At the same time, the expertise of atmospheric scientists in comprehending the chemical properties of various PM components plays a crucial role in elucidating the link between these components and the development of diseases, thereby aiding epidemiologists in causal investigations.

## 6.2 Working with citizen cohorts to establish causal links

The establishment of national biobanks and citizen cohorts is key for investigating the causal links between exposure to PM components and diseases. These cohorts are the gold standard for understanding long-term health effects of environmental factors (Probst-Hensch et al., 2022). They provide evidence where randomized trials are unethical or unfeasible (Peters et al., 2022). Cohorts are critical for approaching a causal understanding of how social, environmental, behavioral, and economic factors promote or hinder health, while also enabling the evaluation of the long-term impacts of public health interventions. They allow studying health trajectories across different ages, providing a life course perspective. As such, they serve as a fundamental pillar for addressing the health effects of PM in the context of other major public health challenges of the 21st century, including population growth, aging societies, urbanization, global warming, digital transformation and increasing social inequalities.

Europe has a longstanding tradition of implementing and maintaining large-scale (>100k participants) and long-term (>20 years) cohorts, including the UK Biobank (Sudlow et al., 2015), Lifelines (Stolk et al., 2008), Constances (Zins et al., 2010), and the German National Cohort (Peters et al., 2022). Innovations in these cohorts include recruitment from birth to old age, implementation of novel eHealth tools, involvements of psychologists and social scientists, and citizen participation during planning and execution to address response rate challenges. Biomaterial collection within these cohorts enables sequencing and in-depth molecular characterization, differentiating between genetic and environmental factors.

A noteworthy addition to national cohorts is the global mortality dataset, maintained by the Global Burden of Disease Collaborative Network, within the Institute for Health Metrics and Evaluation (see Figure 5). Although the dataset is limited to cause-specific mortality, this network has shaped modern epidemiology and allowed the quantification of the global burden to PM mortality (Burnett et al., 2018). Another important dataset is from the multi-country, multi-city network, which provides daily mortality for several locations around the world, ideal the assessment of the short-term PM exposures (Masselot et al., 2022; Liu et al., 2019). Such datasets synergistically complement the causal investigations into PM health effects based on cohort data, providing a global perspective.

Our vision is to integrate detailed knowledge of PM composition with longitudinal personalized medical data of citizen cohorts, to uncover the involvement of specific PM components in disease development and detect early changes resulting from exposure. By working closely with citizen cohorts, epidemiologists and atmospheric scientists will generate compelling evidence for science-to-citizen-to-policy partnership, essential for effecting changes towards a healthier environment. As establishing large-scale cohorts is an immense, multidisciplinary endeavor, it becomes imperative to secure long-term, sustainable funding for study centers, biobanks, and central digital infrastructures dedicated to



data storage and access. Funding should encompass both environmental and health data, recognizing
the integral role of both aspects.

### 6.3 Preventing disease and promoting wellbeing through the mitigation of detrimental PM components

While major attention has been devoted to studying the mortality caused by PM exposure, it is
equally important to consider the impact of PM on morbidity and overall wellbeing. We firmly believe
it is vital to prioritize quality of life and healthy aging over simply extending life expectancy, especially
in high SDI regions. This necessitates a fundamental shift towards primary prevention and the
implementation of drastic changes in health promotion starting at childhood and early adulthood, well
before the onset of diseases. In the case of PM, it is essential to identify and mitigate the specific
components responsible for different diseases, in order to alleviate their impacts on our wellbeing. A
reduction in detrimental PM components will also result in an extension of life expectancy, especially
in low SDI locations.
Dementia serves as a perfect illustration of the major challenges facing our aging society.
Dementia is a severe decline in cognitive function, which considerably affects the wellbeing of older
adults and their families, while imposing substantial costs on public programs. In 2010, approximately
135 million adults were living with dementia worldwide (Prince et al., 2013), resulting in estimated
economic impacts of $600 billion (Wimo et al., 2013). Given the sharp rise in dementia incidence
beyond the age of 75 and our increasingly ageing society, global dementia cases are forecasted to triple
by the year 2050. Recent studies have shown that every 5 µg m$^{-3}$ increase in annual PM concentrations
results in a 13% increased risk of first-time hospital admissions for dementia (Shi et al., 2020), with
elemental carbon and sulfate particles having the strongest effects (Shi et al., 2023). While more
research is necessary to confirm this connection and understand the underlying biological pathways
involved, these studies constitute a first step towards the development of interventions to slow the
trajectory of cognitive decline and ensure the wellbeing of our aging society.
The chemical composition of PM play a key role in mediating its health effects. This inherently
implies that different PM components could potentially be associated to different diseases, possibly
operating through distinct biological pathways in disease development. Building upon the example of
dementia and leveraging established cohorts and biobanks, close collaboration between epidemiologists
and atmospheric scientists becomes evident in identifying the specific PM components responsible for
various diseases and inferring the underlying biological pathways. This collaborative effort is crucial
for mitigating PM impacts on the wellbeing of our society; it combines the expertise of epidemiologists
in understanding disease patterns with the experience of atmospheric scientists in measuring and
modelling air pollution components.

## 7. Conclusions

In the 21st century, we have witnessed a remarkable rise in life expectancy and significant shifts
in global disease patterns, largely attributable to a combination of public health interventions and
advancements in healthcare and healthcare accessibility. Our understanding of risk factors associated
with the early onset and progression of non-communicable diseases has undergone substantial
improvements. Population-based research has played a pivotal role in establishing the influence of
lifestyle determinants on disease outcomes, as well as the intricate role of genetics in disease
progression. Our understanding of long-term environmental exposures to different pollutants and their
contribution to the global burden of disease has significantly improved. It is through this understanding



that we now realize that preventable deaths due to environmental exposures alone range between 9 and
13 million every year (Neira and Prüss-Ustün, 2016; Landrigan et al., 2018), with atmospheric PM
making the largest contribution.
As we continue to deepen our understanding of the impact of environmental factors on public
health, it becomes increasingly evident that solely relying on medical advances will not suffice. We
find ourselves in an era where the returns on investments in high-tech medicine may be diminishing,
jeopardizing the stability of the healthcare system and further exacerbating social inequalities.
Therefore, we strongly advocate for a profound shift in focus towards enhancing quality of life and
healthy aging rather than indiscriminately pursuing life extension at any cost. Central to this paradigm
shift is the need to prioritize early prevention and health promotion strategies, with a particular emphasis
on creating healthy environments. Realizing these strategies will require a combination of large-scale
population health surveillance with precise air quality measurements and modelling, allowing the
determination and mitigation of the main PM components that affect our health. This is only possible
through a close collaboration between atmospheric scientists and epidemiologists, working together to
integrate air pollution exposures with personalized medical data obtained from citizen cohorts.
As an aggressive attempt to promote healthy environments, WHO has set new guidelines to
limit PM concentrations to below 5 $\mu g\ m^{-3}$. Achieving these limits may be challenging for many regions
due to the contribution of natural emissions from wildfires, biogenic species, and desert dust. There is
a need to reconsider how we should be mitigating PM pollution and develop new generation of more
feasible and regionally-specific air quality metrics that focus on detrimental PM components and
exempt non-detrimental or uncontrollable components from guidelines.
Now, we face a pivotal moment where advances in atmospheric science can offer detailed
global air quality maps necessary for establishing epidemiological connections between individual PM
components and health outcomes, thereby, pinpointing the main culprits behind PM health impacts. Our
proposal includes considering elemental carbon, organic aerosols from different sources, ammonium
sulfate, ammonium nitrate, vehicular wear and dust as these key components. Focusing on the
differential toxicity of PM components offers two key advantages. First, it allows for targeted measures
aimed to limit specific health-relevant PM sources. Second, PM chemical composition is intertwined
with other properties that drive PM's health effects, such as solubility, number size distribution and
oxidative potential. Therefore, targeting specific PM components is the most effective approach to
address PM health impacts, enabling targeted measures towards health-relevant PM sources and
considering the properties that drive PM's adverse effects. With the widespread availability of
monitoring data, improved understanding of emissions and their atmospheric aging, and machine
learning integration in atmospheric modelling, the atmospheric science community is now able to
determine the distribution of these components with unprecedented spatial and temporal resolution and
coverage, distinguishing between anthropogenic, controllable and uncontrollable natural emissions.
The use of these distributions in epidemiological analysis will lay the foundation for evidence-based,
targeted interventions that strike the right balance between feasibility and protecting human health.
Routine, widespread availability of high-resolution air quality data in urban centers could have
transformative implications for air quality research, epidemiology, and environmental management.
This valuable data can reveal localized pollution hotspots, offering new opportunities for implementing
targeted pollution control measures. When combined with personal GPS data, it enables comprehensive
personalized exposure analytics, potentially influencing individual behavior. This parallels the way



real-time traffic data currently shape driving patterns at an individual level or how health applications
motivate individuals to engage in active exercise.
By providing open access to global high-resolution pollution maps, atmospheric scientists can
assume a broader societal role in raising public awareness of air pollution and consequently, mitigating
its impacts on public health and environmental equity. These pollution maps empower citizens, local
communities and policy makers with the necessary tools to optimize emission reduction strategies and
sustainable urban planning. This can include the application of targeted measures for limiting the most
important PM sources for health, rather than total PM mass, or shifts in urban land-use design for better
air quality. This wealth of data can be utilized to train models that predict the future evolution of air
pollution sources and its health impacts with climate change, land use change, urban planning,
mitigation strategies and energy policies. Long-term global air quality data are a key cornerstone for
establishing targeted strategies to improve public health and anticipate its future trajectory.
In the process of reevaluating and implementing air quality guidelines, a multidisciplinary
collaborative approach involving atmospheric scientists, climate scientists, epidemiologists, public
health experts, social scientists, policy-makers, and the public is crucial. Therefore, governments must
ensure sustainable funding to foster these collaborations, the returns in terms of lives and costs saved
being increasingly evident. By alleviating the burden of air pollution-related diseases, we will prioritize
the health and wellbeing of individuals and create sustainable and resilient communities.
**Acknowledgments:** We acknowledge the Swiss data science centre (grant Aurora), the Swiss
federal office of environment (FOEN) and the Joint Research Program of the Swiss National Science
Foundation (SNSF grant no. 189883) for their financial support. We also thank the Swiss Agency for
Development and Cooperation (SDC) for financially supporting the Clean Air Project in India (grant
no. 7F-10093.01.04) and the Clean Air China program (grant no. 7F-09802.01.03). We thank the Global
Burden of Disease Collaborative Network and the Institute for Health Metrics and Evaluation (Seattle,
United States) for data in Figure 5 (made available through https://vizhub.healthdata.org/gbd-results/).



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
