# Peer review of "Opinion: how will advances in aerosol science inform our"

_EGUsphere, 2023_

## Author Comment (AC1)

Dear Reviewer,

We thank you for your comments on our opinion article, which significantly improved the paper. Below we provide our point-by-point response to the reviewers' comments. We have added one coauthor 'Petros Vasilakos', who supported addressing the comments and improving the quality of the paper. Comments are in *italic grey typeset*, responses are in regular black typeset, and changes to the manuscript are in blue regular typeset.

**General comments**

**Comment:** *This paper addresses an important topic: how to improve the evidence base to allow the effective targeting of air pollution interventions to improve public health. It includes some useful, relevant information and some interesting examples. However, it reads rather as a series of disjointed sections which aren't drawn together in a clear narrative, and sometimes appear inconsistent. I think the paper would benefit from some redrafting. In particular, it would be helpful to include an introductory section explaining the main purpose of the paper, what it covers, and the authors' view of what it adds to the literature already published on this topic. Some information on how the literature cited was selected would be useful, too.*

**Response:** We thank the reviewer for his comment. We have made the following changes:

- We redrafted the paper to better connect between the sections and reflect the main focus on the importance of including PM chemical composition in epidemiological assessments. These changes could be best seen in blue color throughout the manuscript.
- We have presented existing literature reviews on the topic of each section.
- We have added an introductory section explaining the main purpose of the paper, what it covers, and our view of what it adds to the literature already published on this topic. This section reads as follows:

**1.4 Introductory overview**

In this account, we discuss how the broader atmospheric science community can inform policies and interventions to mitigate sources of PM components that pose risks to human health (Figure 1). We advocate for a foundational shift towards considering PM deferential toxicity in epidemiological health assessments, made possible through improved air quality modelling suitable for exposure assessment, and present the key milestones within aerosol science that, in our view, are necessary for this shift. Section 2 introduces the concept of PM differential toxicity and its potential as an exposure metric. Section 3 critically examines recent advances in modelling tools for estimating fine-scale exposures to specific PM components. In section 4, we identify the type of ambient observations we think are essential for developing and validating exposure models. In section 5, we highlight remaining gaps in our understanding of PM component emissions, their atmospheric transformation and associated health effects and identify research opportunities. Section 6 delves into the need for strong collaboration between research communities to elucidate biological mechanisms underlying the health impacts of specific PM components.

- We have updated the section titles to reflect the common theme of the article centered around PM differential toxicity. The article new table of content is as follows:

**1. Preamble**

**1.1 A brief chronology of air pollution**

**Comment:** *The abstract gives the impression that the paper focuses on the differential toxicity of particulate air pollution; in fact much of the detailed information relates to aspects such as monitoring, modelling, emissions sources, atmospheric chemistry etc. Pulling out some of the conclusions, or specific recommendations for future research, from these sections and including them in the abstract might be useful to the audience. Perhaps the intended focus of the paper is to comment on how improved monitoring and modelling of components / metrics of particulate air pollution could contribute to informing policies and interventions to maximise health improvements? If so, then some of the information included is perhaps not really relevant.*

**Response:** The focus of the paper is on the differential toxicity of PM and how improved monitoring and modelling of PM components could contribute to informing policies and interventions to

maximize health improvements. We have rewritten the abstract to better reflect the paper focus. The abstract reads as follows:

**Abstract.** Air pollution, with high levels of particulate matter (PM), poses the greatest environmental threat to human health, causing an estimated seven million deaths annually and incurring 5% of the global GDP. While PM health impacts are influenced by the toxicity of its individual chemical constituents, the PM mortality burden is solely based on its total mass concentration. This is because of a lack of large-scale, high-resolution PM chemical composition data needed for epidemiological assessments. Identifying which PM constituents are harmful for health has been the 'Holy Grail' of atmospheric science, since the seminal six US cities study that first linked PM to mortality in 1993. Ever since, atmospheric scientists have focused on understanding aerosol composition, emission sources and formation pathways, while longitudinal epidemiological studies needed individual level exposure data, using land use regression models for the prediction of exposures at fine resolutions. In this opinion article, we argue that the time has now come to shift focus towards considering PM chemical composition in epidemiological health assessments, laying the foundation for the development of new regulatory metrics. This shift will enable targeted guidelines and subsequent regulations, prioritizing mitigation efforts against the most harmful anthropogenic emissions. Central to this shift is the availability of global long-term, high resolution PM chemical composition data obtained through field observations and modelling outputs. In the article, we underscore key milestones within aerosol science integral for advancing this foundational shift. Specifically, we examine emerging modelling tools for estimating exposure to individual PM components, present the type of ambient observations needed for model developments, identify key gaps in our fundamental understanding of emissions and their atmospheric transformation and propose a forward cross-disciplinary collaboration between aerosol scientists and epidemiologists to understand the health impacts of individual PM components. We contend that aerosol science has now reached a pivotal moment in elucidating the differential health impacts of PM components, as a first step toward their incorporation into air quality guidelines.

**Specific comments**

**Comment:** *Controllable vs noncontrollable /anthropogenic vs natural sources: There is some inconsistency in the discussion in different sections of the paper regarding PM from natural/uncontrollable sources. The inclusion in the paper of Table 1, outlining evidence gaps and needs related to the health effects of natural PM, suggests that some of the authors consider that these are priorities for research. Other parts of the paper seem to regard these as sources to be dismissed. For example, the paper calls for the exemption from guidelines of components from uncontrollable sources, and recommends collaboration with WHO to achieve this. WHO air quality guidelines are health-based, and do not reflect achievability. Instead, the extent to which sources can be controlled through policy or operational interventions is one of the factors taken into account by legislators when developing national (or regional) regulation or legislation. Whether the lack of control over a source necessarily means that it should be exempted from compliance assessments is a topic of debate - there are health-based reasons that might suggest that it should not – and some discussion of these issues could be included in the paper.*

**Response:** We agree with the reviewer that regulations are health based and do not reflect achievability. Therefore, we have changed the text accordingly. The paper no longer calls for the exemption of uncontrollable components. These revisions can be best identified by the blue font in the text.

We have also added new material to Section 5.1 "Legacy and emerging anthropogenic PM emissions" to have a balanced focus on both anthropogenic and biogenic emissions. This includes completing the list of important anthropogenic emissions, as suggested by reviewer 2 and the addition of the new table 2 on the gaps related to anthropogenic emissions. The revised text in section 5.1 reads as follows:

Anthropogenic emissions remain a predominant source of primary and secondary PM, posing a critical scientific and policy challenge in identifying the most harmful components to human health. Existing reviews have compiled epidemiological and toxicological evidence linking specific emissions to health endpoints (Wyzga and Rohr, 2015; Adams et al., 2015; Rohr and Wyzga, 2012; Yang et al., 2019). While ample literature covers short-term effects, especially through measurements at few stations, longitudinal epidemiological studies investigating the effect of PM chemical composition on chronic health outcomes are relatively scarce. Despite inconsistencies across studies, elemental carbon, organic aerosols, sulfate and metals have been consistently associated with increasing risks of cardiovascular and respiratory mortality and hospitalization (Chen et al., 2018a; Yang et al., 2019; Masselot et al., 2022; Wang et al., 2022; Wyzga and Rohr, 2015; Adams et al., 2015; Rohr and Wyzga, 2012; Badaloni et al., 2017; Wang et al., 2017).

We believe that a major limitation in establishing robust epidemiological associations with specific PM components has been the correlation between these components and with other pollutants (e.g. $O_3$ and $NO_X$). Therefore, we call for improved high-resolution large scale chemically detailed exposure models that will offer the necessary variability for overcoming limitations related to correlations. Moreover, we advocate for the continual development of epidemiological multi-component methods that estimates the joint health impacts of PM components, instead of isolating the effect of individual ones. In this section, we will focus on major anthropogenic emissions, including fossil fuel emissions, non-exhaust on-road emissions, volatile chemical products (VCPs), and residential biomass burning (Table 2).

[revised manuscript text omitted]

Fundamental studies of biomass smoke aging. | Fine resolution modelling of biomass burning emissions.

Implementing biomass burning aging mechanisms in models.

. |

**Comment:** *As the paper illustrates in Figure 6, categorisation of sources of PM as controllable or not controllable is not straightforward. Land-use and human activities can influence the emissions of biogenic VOCs and the likelihood of wildfires, for example. And the WHO good practice statement on particles originating from sand and dust storms (SDS) (in the WHO 2021 AQG document) includes measures that can be implemented to mitigate exposure. This distinction between natural/uncontrollable and anthropogenic/controllable emissions could therefore be discussed in a more nuanced way in the paper.*

**Response:** We have modified the discussion about the distinction between natural/uncontrollable and anthropogenic/controllable emissions as follows:

In the introduction of Section 5:

Human activities have profoundly altered the earth's environment, impacting emissions, atmospheric composition, global temperatures, and land cover. In Figure 6, we categorize the complex anthropogenic effects on PM composition into four broad classes:

- **Direct emissions:** encompassing anthropogenic PM and PM precursors directly released into the atmosphere.
- **Land-use changes:** including changes in urban infrastructure, green initiatives, deforestation/forest management, and agricultural practices, affecting emissions and their accumulation patterns.
- **Direct effects of anthropogenic emissions on the chemistry of natural PM:** whereby pollutants from human activities react with biogenic emissions leading to PM formation.
- **Indirect perturbation of natural PM:** through anthropogenic emissions that impact natural ecosystems, such as global warming, increased $CO_2$ concentrations, shifts in vegetation patterns, or desertification.

This section addresses existing gaps in understanding anthropogenic emissions, their atmospheric transformation, and their direct and indirect influence on natural PM. It is crucial for the atmospheric science community to approach these gaps from a mechanistic standpoint and incorporate them into models to accurately quantify the anthropogenic impacts on PM composition and thereby health effects. In section 5.1, we discuss anthropogenic PM sources that hold relevance for public health, while in section 5.2, we examine the future trajectory of the natural PM background and its interactions with anthropogenic activities.

In the introduction of Section 5.2:

With the increasing regulations on anthropogenic emissions, the contribution of natural emissions, including biogenic volatile organic compounds (BVOCs), wildfires and desert dust, will gain prominence (Figure 4). While these emissions stem from natural ecosystems, they are also significantly perturbed by anthropogenic activities, as illustrated in Figure 6. The traditional picture that distinguishes biogenic and anthropogenic sources obscures human impacts on ostensibly natural systems. Anthropogenic effects on natural PM can be either direct, through the alteration of atmospheric reactivity, or indirect, through feedback mechanisms triggered by changes to the biosphere. We need to understand these effects quantitatively to devise best practices to mitigate their impacts. For example, WHO good practice statement on particles originating from sand and dust storms (SDS in the WHO 2021 air quality guideline document) includes measures that can be implemented to mitigate exposure. In this section, we discuss the human influence on natural PM concentrations, chemical composition, and future trends (Table 3).

In the conclusion of Section 5:

While anthropogenic emissions are destined to decrease, natural emissions will most likely increase. Part of this increase can be controllable through reducing anthropogenic emissions and managing land-use. The atmospheric science community is now ready to provide the field measurements, laboratory observations and model outputs needed to quantify the contribution of anthropogenic, and controllable natural emissions globally and predict their evolution with global

changes. This data will form the foundation for understanding the toxicity of anthropogenic PM sources and determining the natural PM background in different regions.

**Comment:** *Figure 4: The source of the data underpinning the illustration (in Figure 4) of the contribution of natural components to total PM, and how it varies with PM mass concentration, is not clear. The source of this data should be given, so that readers can access it. (Some locations / regions with high total PM also have high contributions of "natural" PM – arising from sources such as wind-blown desert dust, or wildfires – this doesn't seem to be reflected in the discussion, or in the figure.)*

**Response:** We have added the source of the data illustrated in the caption of Figure 4, as follows:

For illustration, we have chosen a natural background concentration of 5 µg m$^{-3}$, representing the level to which 50% of the global population would be exposed if all anthropogenic emissions were eliminated (Pai et al., 2022).

**Comment:** ***Targeting interventions: based on toxicity or source contribution?:*** *it is unclear whether the authors' overall focus is on identifying PM components that are most detrimental to health or identifying local sources that are major contributors to PM mass concentration and should therefore be targeted. Both are important, and both are discussed – but separately. A summary of the different ways in which atmospheric science, monitoring and modelling can inform policy-making and operational decisions would be useful to tie these different aspects together.*

**Response:** The overall focus of the paper is on PM differential toxicity and the identification and abatement of detrimental PM components. Indeed, some of these components are also major PM contributors and targeting them would result in a reduction in total PM mass concentration. The revised paper now presents more clearly the different ways in which atmospheric science, monitoring and modelling can inform policy-making and operational decisions. In addition, we have rewritten the first part of the conclusion, adding a summary of developments in atmospheric science needed to inform policy -making and operational decisions. The modified conclusion section reads as follows:

In the 21st century, we have witnessed a remarkable rise in life expectancy and shifts in global disease patterns, attributable to a combination of public health interventions and advancements in healthcare and healthcare accessibility. Yet, ten million deaths attributable to environmental exposures can still be preventable every year (Neira and Prüss-Ustün, 2016; Landrigan et al., 2018), highlighting the need for proactive measures. Relying solely on high-tech medical interventions for managing disease progression may exacerbate existing social inequalities within healthcare systems and yield diminishing returns. Therefore, we advocate to shift towards enhancing quality of life and promoting healthy aging through early prevention and the creation of healthy environments for all. Our vision for realizing this goal is a close collaboration between atmospheric scientists and epidemiologists, to integrate chemically detailed global air quality data with large-scale personalized medical information from citizen cohorts. The provision of global PM composition maps will require the atmospheric science community to (1) develop spatially and chemically detailed exposure models, (2) provide long-term time-series of PM chemical composition from monitoring networks, (3) map pollution hot-spots through mobile measurements, (4) understand emerging anthropogenic emissions and their chemical transformation, especially in heavily polluted areas like China and India, and (5) understand the future evolution of natural emissions with climate and land-use changes.

As an aggressive attempt to promote healthy environments, WHO has set new guidelines to limit PM concentrations to below 5 µg m$^{-3}$. Achieving these limits may be challenging for many

regions due to the contribution of natural emissions from wildfires, biogenic species, and desert dust. Concurrently, scientific consensus underscores the critical role of PM chemical composition in influencing associated health effects, necessitating a revaluation of how we should be mitigating PM pollution and the development of new generation of air quality metrics focusing on detrimental PM components. Focusing on the PM differential toxicity offers two key advantages. First, it allows for targeted measures aimed to limit specific health-relevant PM sources. Second, PM chemical composition is intertwined with other properties that may also drive PM's health effects, such as solubility, number size distribution and oxidative potential. Atmospheric science has reached a pivotal moment to provide detailed global air quality maps, at a sufficiently fine resolution, supporting epidemiological studies to determine the differential toxicity of PM components, crucial for integrating PM chemical composition into regulatory frameworks, informing targeted policy-making and operational decisions.

**Comment:** *Indoor air: There is inconsistency in different sections of the paper in the way that indoor pollutants are addressed. Early in the paper, the authors suggest that indoor air pollution "should be treated as a separate risk factor distinct from outdoor air pollution, akin to contaminated water". The reasoning which led the authors to this view is not clear: is it because different policies are needed to address emissions from indoor and outdoor sources, for example? Conversely, later in the paper, considerable emphasis is put on volatile chemical products (VCPs - including cleaning agents and personal care products, which are used indoors) as sources of outdoor organic aerosol. This inconsistency should be addressed.*

**Response:** Our reasoning on treating indoor and outdoor sources separately is because (1) different regulatory frameworks are needed to address emissions from indoor and outdoor sources, (2) these sources are often distinct, and (3) they require different control measures. Indeed, some indoor sources are also important sources of outdoor pollution such as VCPs.

We had revised Section 2 to better reflect our view on the distinction between indoor and outdoor sources:

In epidemiological analyses, outdoor PM concentrations at residences are commonly used as proxies for exposure. While there is evidence supporting this approach, its applicability across different settings requires further investigation (Wei et al., 2023). As we spend most of our time indoors and new buildings are increasingly airtight for energy saving, outdoor concentrations may not reflect indoor levels (Schweizer et al., 2007). While indoor emissions, primarily from cooking (Klein et al., 2019) and smoking (Hyland et al., 2008), may influence health, they represent a separate risk factor distinct from outdoor air pollution, akin to contaminated water. This is because (1) different regulatory frameworks are needed to address emissions from indoor and outdoor sources, (2) these sources are often distinct, and (3) they require different control measures. Unlike outdoor air pollution, which often requires collective and regulatory abatement strategies to control emissions, indoor air pollution can be more effectively managed at the individual or household level, by improving ventilation and eliminating or reducing indoor sources. In the absence of indoor emissions, indoor concentrations are 30 to 70% lower than outdoors (Chen and Zhao, 2011) due to variability in infiltration rates. Moreover, exposures can also be influenced by outdoor pollution in other settings, such as workplaces and during commuting, where we spend a large fraction of our time. Health data from citizen cohorts often include questionnaires that offer valuable insights into indoor infiltration rates, workplace conditions and individual's mobility. While we consider outdoor concentrations at residence to be a reasonable proxy of exposure to outdoor pollution, integrating such information can

help refining exposure estimations. First, however, the issue of downscaling air quality models to finer resolutions must be tackled.

We had also revised Section 5.1 mentioning that a large fraction of VCPs may come from indoor sources:

With the drastic reduction of on-road transportation emissions, VCPs, which are partly from indoor emissions, have emerged as one of the largest sources of outdoor urban organic emissions in US and European cities, modulating urban chemistry (Coggon et al., 2021; Gkatzelis et al., 2021; Mcdonald et al., 2018).

**Comment:** *Differential toxicity: Section 5.1 "Health effects of anthropogenic PM emissions" includes "Our review reveals mixed results regarding the differential health effects associated with different anthropogenic PM components". How was this review undertaken? What search terms and literature sources were used? Were recent reports which have reviewed the evidence related to the differential toxicity of ambient PM consulted? [Examples include (USEPA PM ISA, 2019; ANSES, 2019; COMEAP, 2022) and the HEI NPACT initiative.]*

**Response:** The paper is not meant as a systematic review of the toxicological and epidemiological evidence on PM differential toxicity, as previous reports, which are the basis for WHO regulations, have already offered a much more thorough overview. Therefore, we changed the title of Section 5.1 to "Legacy and emerging anthropogenic PM sources". Furthermore, we have omitted from Table 2 and Table 3 the last column: level of scientific understanding, which deserves a dedicate review. Following the reviewer comment we have cited the recent reports which have reviewed the evidence related to the differential toxicity of ambient PM. This section reads as follows:

Anthropogenic emissions remain a predominant source of primary and secondary PM, posing a critical scientific and policy challenge in identifying the most harmful components to human health. Existing reviews have compiled epidemiological and toxicological evidence linking specific emissions to health endpoints (Wyzga and Rohr, 2015; Adams et al., 2015; Rohr and Wyzga, 2012; Yang et al., 2019)(Morton Lippmann Lung, 2023 #3276). While ample literature covers short-term effects, especially through measurements at few stations, longitudinal epidemiological studies investigating the effect of PM chemical composition on chronic health outcomes are relatively scarce. Despite inconsistencies across studies, elemental carbon, organic aerosols, sulfate and metals have been consistently associated with increasing risks of cardiovascular and respiratory mortality and hospitalization (Chen et al., 2018a; Yang et al., 2019; Masselot et al., 2022; Wang et al., 2022; Wyzga and Rohr, 2015; Adams et al., 2015; Rohr and Wyzga, 2012; Badaloni et al., 2017; Wang et al., 2017).

**Comment:** *As the paper notes, epidemiology using chemical speciation data will be key to investigating which components of PM might be most health-relevant. However, there will be limitations to how far epidemiology, alone, can address this question. If differential toxicity is to be a main focus of the paper (as suggested by the abstract) it would benefit from more discussion of these limitations. The authors note that confounding might occur because of the strong correlation between various PM components. Confounding by other co-emitted or co-located pollutants (eg NO₂, VOCs, SO₂) is likely an equally important issue, which should be mentioned. Such limitations suggest a need*

*for experimental toxicological data (in vitro and/or in vivo) to inform considerations of differential toxicity.*

**Response:** We agree with the reviewer that the strong correlation between various PM components and with other pollutants is a major limitation of epidemiology. However, we contend that long-term, large scale and high-resolution data would help overcoming the problem of correlation. We have clarified our viewpoint in section 5.1 as follows:

We believe that the principal challenge in establishing robust epidemiological associations with specific PM components lies in their correlation with other pollutants, such as other PM components, $O_3$ and $NO_X$. Therefore, we call for improved high-resolution large scale chemically detailed exposure models that will offer the necessary variability for overcoming limitations related to correlations. Moreover, we advocate for the continual development of epidemiological multi-component methods that estimates the joint health impacts of PM components, instead of isolating the effect of individual ones. In this section, we will focus on major anthropogenic emissions, including fossil fuel emissions, non-exhaust on-road emissions, volatile chemical products (VCPs), and residential biomass burning (Table 2).

**Comment:** *Attributable mortality: I would recommend making it clearer that all air pollution mortality burden figures are estimates, and are dependent upon the underpinning assumptions and data used (estimated pollution concentrations, exposure-response functions, counterfactuals etc). The approaches may differ between the different estimates quoted. I would also suggest use of a term such as "attributable deaths" or "an effect equivalent to x deaths" or similar, rather than "premature" deaths: in public health, "premature deaths" is often used refer to deaths in those aged less than 75 years old.*

**Response:** Based on the reviewer comment, we have replaced the term premature deaths by attributable deaths or estimated deaths. These modifications are best seen by the blue font in the updated version of the manuscript.

*Dosimetry of PM within the lung, translocation and causation of health effects:* discussion of these aspects could be more nuanced (for example, only a very small proportion of even nano-sized particles are understood to enter the blood stream). But I don't think that this is a main focus of the paper, so an alternative might be to scale these sections back.

**Response:** Based on the reviewer comment, we have significantly scaled section 6 back. The focus of this section is on the collaboration between atmospheric scientists and epidemiologists. The revised version can be seen in the main text.

**Technical corrections**

**Comment:** *Line 32: "about 400 before our era" is unclear. "400 BCE" is more commonly used*

**Response:** modified to 400 BCE.

**Comment:** *Line 154: "To quantify the health impacts of PM, we currently rely on dose-response relationships that link **cause-specific** mortality" : many authoritative organisations use all-cause/natural cause mortality as the basis of estimates, rather than cause-specific mortality.*

**Response:** We removed cause-specific in the updated version of the manuscript.

**Comment:** *Line 183 "insoluble particles, such as asbestos or elemental carbon, can **bioaccumulate** and lead to chronic inflammation". The more correct term is "biopersistence" or*

*similar (bioaccumulation is more usually used for accumulation of chemicals within food chains, for example bioaccumulation of dioxins in fish species such as salmon).*

**Response:** We have modified the sentence as follows:

'whereas insoluble particles like asbestos or elemental carbon are biopersistent in the body, leading to chronic inflammation.'

**Comment:** *Line 869 "WHO has set new guidelines to limit **PM** concentrations to below 5 μg m$^{-3}$." This should specify PM$_{2.5}$*

**Response:** We have specified that this is for PM$_{2.5}$.

**Comment:** *Footnote 1: For this audience, I think the formal definition of PM$_{10}$ and PM$_{2.5}$ should also be included.*

**Response:** We have already defined PM$_{10}$ and PM$_{2.5}$ in the footnote as: Particulate matter with a size lower than 2.5 and 10 μm, respectively.

**Comment:** *Some of the referencing needs to be checked. For example:*

- *Line 102 "WHO, has recently updated its air quality guidelines to propose a much more stringent limit value of 5 μg m$^{-3}$ (Who)" – this reference is not listed*
- *Line 246 (Pope **Iii** et al., 2002)*

**Response:** We have checked and adjusted the references.

---

## Author Comment (AC2)

Dear Reviewer 2,

We thank you for your constructive feedbacks, which improved the paper. Below we provide our point-by-point response to the reviewers' comments. We have added one coauthor 'Petros Vasilakos', who supported addressing the reviewers' comments and improving the quality of the paper. Comments are in *italic grey typeset*, responses are in regular black typeset, and changes to the manuscript are in blue regular typeset.

**Comment:** *This long manuscript proposed opinions about how are advances in aerosol science informing understanding of the health impacts of outdoor particulate matter pollution. It's a very detailed review paper systematically introduced the development of aerosol pollution and corresponding control demands, summarized the globally monitoring or modelling and discussed the roles of PM components on human health. Some improving comments are suggested for considering as follows:*

**Response:** We thank the reviewer for their comments, which we address below.

***General Comments:***

*Key objective of this article was to introduce the concept of using specific PM components as metrics for health assessments in addition to total PM mass, and for reevaluation of air quality guidelines. However, the health effects of PM constituents had been widely cognized and investigated either by toxicology or epidemiology studies, so the known and unknown of this issue might be key contents and suggested in this review. There is still road for connecting specific PM components independently with health effects by reliable epidemiological evidences and toxicological mechanisms clearly, qualitatively and quantitatively.*

**Response:** The reviewer is correct that previous epidemiological studies have considered the effect of PM composition on PM health effect. However, these studies mainly focus on PM acute effects combining daily mortality and morbidity data, typically on a city scale, with measured or modelled PM composition typically at a background site. However, most of PM induced mortality is caused by chronic exposure, the quantification of which requires PM chemical composition data ideally at address level. Such data have been until very recently rarely available, especially for large scales. Hence, connections between chronic exposures and PM chemical composition have rarely been established. In the corrected version of the manuscript, we have added an explanation in this regards, which reads as follows:

**3. Modelling exposures to individual PM components**

The investigation of acute health effects requires the time-series analysis of daily exposures to specific components, typically obtained at an urban background site through long-term measurements (> 3 years) or modelling outputs. By contrast, longitudinal epidemiological studies of chronic diseases require long-term exposures determined at high spatial resolution – ideally at address level. Because high resolution PM composition data are scarce, existing epidemiological analyses considering PM chemical composition have mainly focused on acute effects, while health effects resulting from chronic exposure to individual PM components have rarely been assessed. In this section, we define the state-of-the-art in modelling PM exposure and discuss how recent advances in modelling PM chemical composition can help informing our understanding of PM differential toxicity.

**Comment:** *Moreover, PMs are chemical and biological mixture, their combined health effects result in the total PM mass exposure effects, surely not only additive by individual components.*

**Response:** The reviewer is correct. We have added this information in Section 5.1, as follows:

Moreover, we advocate for the continual development of epidemiological multi-component methods that estimates the joint health impacts of PM components, instead of isolating the effect of individual ones.

And Section 6.2, as follows:

Working together will advance understanding of the involvement of specific PM components or combination of components in disease development and the detection of early changes resulting from exposure.

**Comment:** *Furthermore, the aerosol pollution varied spatially and temporally, the moving personal exposure also varied spatial-temporally, how could the PM monitoring serve the health risk assessments more helpful?*

**Response:** We consider outdoor concentrations at residence place to be a reasonable proxy of exposure to outdoor pollution. To determine the exposure at residence place, PM composition data from multiple fixed sites can be a suitable strategy – this is what is used for PM. However, we consider it is also important to integrate personal mobility data to refine exposure estimations. In section 2.2, we provide our opinion how information about human mobility can be considered when estimating human exposures. The section reads as follows:

Moreover, exposures can also be influenced by outdoor pollution in other settings, such as workplaces and during commuting, where we spend a large fraction of our time. Health data from citizen cohorts often include questionnaires that offer valuable insights into indoor infiltration rates, workplace conditions and individual's mobility. While we consider outdoor concentrations at residence to be a reasonable proxy of exposure to outdoor pollution, integrating such information can help refining exposure estimations.

In section 4.3, we also mention how urban mapping can help refining exposure estimation, including the effect of mobility. The section reads as follows:

Street-level PM composition data can enhance, challenge, or confirm various air quality datasets used to retrieve PM differential toxicity, such as CTM outputs, land-use regression predictions, and remotely sensed observations. This refinement can also aid addressing the effect of human mobility in epidemiological studies (Zeger et al., 2000).

**Comment:** *Since there are both primary and secondary aerosols from both natural and anthropogenic sources, not all components are known and could be monitored in the complicated atmospheric PMs. All these facts should be considered in current discussion.*

**Response:** We believe that the eight components we have suggested are directly measurable or traceable. Their inclusion into epidemiological assessments could be a first step towards considering the PM differential toxicity in regulations. These eight fractions include: organic aerosol, elemental carbon, sulfate, nitrate, ammonium, sea-salt, brake-wear copper, and dust. All components have a dominant and known source except for the organic fraction, which can originate from primary and secondary, natural and anthropogenic sources. While these organic aerosol classes cannot be directly measured, they might be retrieved through receptor modelling based on spectrometric measurements or chemical transport modelling. This is discussed in section 2.1.

**Comment:** *In the key section of PM components, the list of known PM components to be monitored could be showed. In the section of PM sources, not all sources were covered and suggested to complete.*

**Response:** In section 2.1, we included Table 1 to represent PM components that we suggest monitoring. The new table 1 is shown below.

**Table 1:** PM chemical components suggested to be monitored and modelled for integration into epidemiological assessments and determination of PM differential toxicity. Components' physical properties that are important determinant of health effects are shown, including size, morphology, and solubility. Components major sources are also shown. a), b), and c) Transmission electron microscopic images of organic aerosol, elemental carbon aggregates and sulfate adapted from (Li et al., 2011). d) Scanning electron microscopic image of a fresh sea-salt particle adapted from (Li et al., 2016b). e) Scanning electron microscopic image of a coarse brake ware particle adapted from (Kukutschová and Filip, 2018). f) Transmission electron microscopic image of a mineral dust particle adapted from (Xu et al., 2021).

| Component | Size | Morphology | Solubility | Source |
|---|---|---|---|---|
| Organic aerosol | Fine | a) 300nm | Moderately soluble for POA

Soluble for SOA | Natural and anthropogenic, primary and secondary |
| Elemental carbon | Fine | b) 300nm | Insoluble | Biomass and fossil fuel combustion |
| Sulfate | Fine | c) 300nm | Soluble | Aqueous (65%) and gas phase OH (35%) oxidation of $SO_2$ from natural marine emissions (15%) and anthropogenic emissions from electricity generation and industries (85%) (John H. Seinfeld, 2016) |
| Nitrate | Fine | | Soluble | Oxidation of $NO_X$ emissions mainly from traffic exhaust |
| Ammonium | Fine | | Soluble | Condensation of gas-phase ammonia mainly from agriculture emissions producing ammonium sulfate and nitrate |
| Sea salt | Coarse | d) 1μm | Soluble | Natural marine emissions through bursting bubbles at the air-sea interface |
| Brake wear (Cu) | Coarse | e) 5μm | Depending on the element | Brake pads |
| Mineral dust (Al, Si, Ti, Fe) | Coarse | f) 3μm | Depending on the element and atmospheric age | Mainly natural wind-blown dust |

In section 5.1, we have discussed a more exhaustive list of sources and have include Table 2 to represent the gaps. This is shown below:

Anthropogenic emissions remain a predominant source of primary and secondary PM, posing a critical scientific and policy challenge in identifying the most harmful components to human health. Existing reviews have compiled epidemiological and toxicological evidence linking specific emissions to health endpoints (Wyzga and Rohr, 2015; Adams et al., 2015; Rohr and Wyzga, 2012; Yang et al., 2019)(Morton Lippmann Lung, 2023 #3276). While ample literature covers short-term effects, especially through measurements at few stations, longitudinal epidemiological studies investigating the effect of PM chemical composition on chronic health outcomes are relatively scarce. Despite inconsistencies across studies, elemental carbon, organic aerosols, sulfate and metals have been consistently associated with increasing risks of cardiovascular and respiratory mortality and hospitalization (Chen et al., 2018a; Yang et al., 2019; Masselot et al., 2022; Wang et al., 2022; Wyzga and Rohr, 2015; Adams et al., 2015; Rohr and Wyzga, 2012; Badaloni et al., 2017; Wang et al., 2017).

We believe that the principal challenge in establishing robust epidemiological associations with specific PM components lies in their correlation with other pollutants, such as other PM components, $O_3$ and $NO_X$. Therefore, we call for improved high-resolution large scale chemically detailed exposure models that will offer the necessary variability for overcoming limitations related to correlations. Moreover, we advocate for the continual development of epidemiological multi-component methods that estimates the joint health impacts of PM components, instead of isolating the effect of individual ones. In this section, we will focus on major anthropogenic emissions, including fossil fuel emissions, non-exhaust on-road emissions, volatile chemical products (VCPs), and residential biomass burning (Table 2).

[revised manuscript text omitted]

**Comment:** *Particle size is a very important parameter influencing PM health risks, which is also related to components and sources, but was not considered much in this paper.*

**Response:** In Table 1 (please see previous comment), we have now listed PM size of different chemical fractions as an important determinant of health effects. We have also specified the fraction of PM in WHO regulations.

**Comment:** *Finally, a question is suggested for consideration: How to control the specific harmful components in the particles selectively?*

**Response:** There exist strong correlations between PM chemical composition and sources (Table 1), for example sulfate with electricity generation or nitrate with traffic exhaust. We have added the following sentence in Section 2.1

The classification of aerosols based on their chemical composition will also establish a direct link to aerosol sources (Table 1), offering policy makers effective and operational strategies for selectively mitigating the most important PM sources for health.

*Minor comments:*

**Comment:** *Line 102: Who should be WHO. Check overall similar typos.*

**Response:** Done in the updated version of the manuscript.

**Comment:** *More figures than words are suggested to show the opinions.*

**Response:** We have added 2 new tables to better illustrate our opinions (Table 1 and Table 2, shown above).